# Extended anharmonic collapse of phonon dispersions in SnS and SnSe

T. Lanigan-Atkins [1,6], S. Yang[1,6], J. L. Niedziela [2], D. Bansal[1], A. F. May [2], A. A. Puretzky[3], J. Y. Y. Lin[4], D. M. Pajerowski[4], T. Hong [4], S. Chi [4], G. Ehlers [4] & O. Delaire [1,5]

The lattice dynamics and high-temperature structural transition in SnS and SnSe are investigated via inelastic neutron scattering, high-resolution Raman spectroscopy and anharmonic first-principles simulations. We uncover a spectacular, extreme softening and reconstruction of an entire manifold of low-energy acoustic and optic branches across a structural transition, reflecting strong directionality in bonding strength and anharmonicity. Further, our results solve a prior controversy by revealing the soft-mode mechanism of the phase transition that impacts thermal transport and thermoelectric efficiency. Our simulations of anharmonic phonon renormalization go beyond low-order perturbation theory and capture these striking effects, showing that the large phonon shifts directly affect the thermal conductivity by altering both the phonon scattering phase space and the group velocities. These results provide a detailed microscopic understanding of phase stability and thermal transport in technologically important materials, providing further insights on ways to control phonon propagation in thermoelectrics, photovoltaics, and other materials requiring thermal management.

[1] Department of Mechanical Engineering and Materials Science, Duke University, Durham, NC, USA. [2] Materials Science and Technology Division, Oak Ridge National Laboratory, Oak Ridge, TN, USA. [3] Center for Nanophase Materials Sciences, Oak Ridge National Laboratory, Oak Ridge, TN, USA. [4] Neutron Scattering Division, Oak Ridge National Laboratory, Oak Ridge, TN, USA. [5] Department of Physics, Duke University, Durham, NC, USA. [6] These authors contributed equally: T. Lanigan-Atkins, S. Yang. ✉email: olivier.delaire@duke.edu

mproving renewable energy technologies such as photovoltaics and thermoelectrics requires a deeper understanding of the atomistic processes underlying their energy conversion and thermal behavior. In particular, fundamental studies of how anharmonicity in the interatomic potential energy surface impacts both phase stability and thermal transport are required. SnS is a candidate for both solar cells and thermoelectric devices. It exhibits strong anharmonicity and a structural phase transition at high temperature, making it ideal to investigate the fundamental effects of anharmonicity[1–4]. SnS is particularly suitable as an absorber in solar cells due to its high optical absorption coefficient and direct bandgap $E_g \simeq 1.3\,eV$ (ref. [5]), and has recently demonstrated fast improvements in thermoelectric efficiency[6,7]. Its bulk electronic structure is also attracting strong interest for valleytronics applications[8]. The sister compound SnSe has been one of the most intensely studied thermoelectric materials in recent years and has set records for thermoelectric conversion efficiency, in part because of its ultralow thermal conductivity[4,9]. Critically, the lattice dynamics of both systems remain insufficiently understood, especially at high temperatures where their crystal structure undergoes a structural phase transition known to be associated with very strong anharmonicity but whose underlying mechanism remains controversial.

Using high-resolution inelastic neutron scattering (INS) on single-crystals, we chart the evolution of phonons in SnS from 150 K to 1050 K and clearly reveal the soft-mode nature of the Pnma–Cmcm phase transition. Further, we find that the transition is accompanied by a striking extensive anharmonic collapse of the transverse acoustic (TA) and transverse optical (TO) branches along extended portions of the dispersion, as opposed to a proposed condensation at a single high-symmetry wavevector $\mathbf{Q} = Y$ (refs. [10,11]). We also succeed in mapping dispersions across the phase transition in SnSe, extending our previous results that were limited to Pnma[12]. Supplementing our INS data with high-resolution Raman spectroscopy and anharmonic first-principles simulations, we demonstrate that drastic, anisotropic, and mode-dependent anharmonic phonon renormalizations occur near a phase transition, and that these impact thermal conductivity, by altering both group velocities and the phase space for phonon-phonon interactions.

SnS and SnSe are chemically and structurally similar. Both compounds crystallize in a Pnma phase (space group 62) at low temperature, and transform to a higher symmetry Cmcm structure (space group 63) when heated above $T_C = 807\,K$ (SnSe) or 878–887 K (SnS)[10,11,13,14]. The Pnma and Cmcm structures are both orthorhombic and are illustrated in Fig. 1, showing the low-temperature distortion of the Sn coordination polyhedron from displacements along [001]. Specifically, Sn and Se atoms exhibit a gradual shift along $c$ in the Pnma phase, which breaks a mirror symmetry of the Cmcm phase[10,11,13,14]. Importantly, the thermoelectric figure-of-merit is reported to peak above 700 K near the structural transition[6,9], yet the mechanism of the transition has remained the matter of debate. Based on the continuous evolution of lattice parameters and bond lengths measured with neutron diffraction, and the group–subgroup symmetry relation of Cmcm and Pnma, Chattopadhyay et al. characterized the transition as second-order, and predicted a soft phonon mode distortion at the $Y$ zone-boundary point of the Cmcm phase, re-emerging as an $A_g$ zone-center optic phonon mode in the Pnma phase[10,11]. But this was never directly confirmed, as no phonon dispersions have been reported for SnS at all, while only limited low-resolution INS measurements were performed in SnSe for $T$ reaching near $T_C$ (refs. [12,15,16]). Conversely, some studies have found evidence that the transition has some degree of first order character, and could involve a possible two-step mechanism[14,17] but this model remains controversial, as recent anharmonic phonon simulations are consistent with a second-order behavior[18].

Because of strongly anharmonic bonding, both compounds exhibit very low thermal conductivities (single crystal $\kappa$ at 300 K —SnSe: 1.6 (ref. [19]), SnS: 2.42 (this work) $Wm^{-1}K^{-1}$ with both compounds reaching $\kappa < 1\,Wm^{-1}K^{-1}$ at high temperature), which motivated numerous studies[12,15,17,20,21]. Despite huge interest in the low $\kappa$ for thermoelectrics, very little experimental information is available on phonons at high-temperature, especially in the Cmcm phase. Hence, the phonon behavior across the phase transition has remained unclear. High-temperature Raman data were recently published for SnSe but were limited to modes above 40 cm$^{-1}$ (ref. [22]), thus excluding low-energy optical modes. At 300 K, the energy of the Raman-active soft mode predicted by Chattopadhay et al.[10] is 33 cm$^{-1}$ (SnSe) and 40 cm$^{-1}$ (SnS)[23], and further softens on heating, requiring measurements close to the laser line.

INS is the most powerful technique to map mode-resolved linewidths and group velocities ($v_g$) across the Brillouin zone, thus providing crucial insights into the thermal conductivity. Indeed, our previous INS studies on SnSe at low temperature uncovered a strong anharmonicity and coupling between the phonons and the electronic structure[12] but were limited to the low-temperature phase of SnSe (ref. [12]). Mapping phonon dispersions at $T > 800\,K$ in either SnS or SnSe has proved challenging owing to sample sublimation, but the current measurements circumvented this difficulty (see "Methods") and achieved a comprehensive view of the lattice dynamics across the structural phase transition into the Cmcm phase.

## Results

**Diffraction/phase transition.** The Pnma–Cmcm structural distortion and our single-crystal neutron diffraction results on SnS are shown in Fig. 1. Pnma notation is used throughout to describe the (hkl) peak indices, corresponding to conventional orthorhombic cells in Fig. 1a, e. Owing to the doubling of the primitive cell volume in Pnma, superstructure Bragg peaks occur at zone-boundary $Y$ points of the primitive Cmcm cell, for example at (201) and (102) in the low-temperature phase, and disappear above $T_C$ (~880 K) (Fig. 1b, f). Panels 1c, d, g show that the intensities of superstructure Bragg peaks (102) and (201) continuously decrease on warming until full suppression at $T_C$ and above. Conversely, (002) and (101) are not superlattice peaks and their intensities increase with temperature. The continuous evolution of the peak intensities and lattice parameters (Fig. 1h) are supportive of a second-order (or weakly first-order) transition scenario. A strong anisotropy is also observed, with the c-axis contracting, while a and b exhibit a normal expansion on heating, see Fig. 1h and prior studies[11,13,24]. The thermal contraction along $c$ in the Pnma phase was partially explained by a quasi-harmonic (QH) analysis of the free-energy, yet the stronger lattice parameter variations close to $T_C$ pointed to strong intrinsic anharmonicity in this regime[25]. Above $T_C$, the change in lattice parameters becomes weaker. From our complementary heat capacity measurements (Supplementary Fig. 18), the phase transition in SnS occurs at 875 K which is consistent with our neutron data and prior results[24].

**Phonon dispersions.** A drastic change in phonon dispersions is observed across the structural phase transition in both SnS (Fig. 2) and SnSe (Supplementary Figs. 1, 2). High energy resolution was required to accurately map low energy modes in the dynamical susceptibility $\chi''(\mathbf{Q},E)$, and, therefore, we used the cold neutron chopper spectrometer (CNCS) at the spallation neutron source (SNS) and the cold triple-axis spectrometer (CTAX) at the

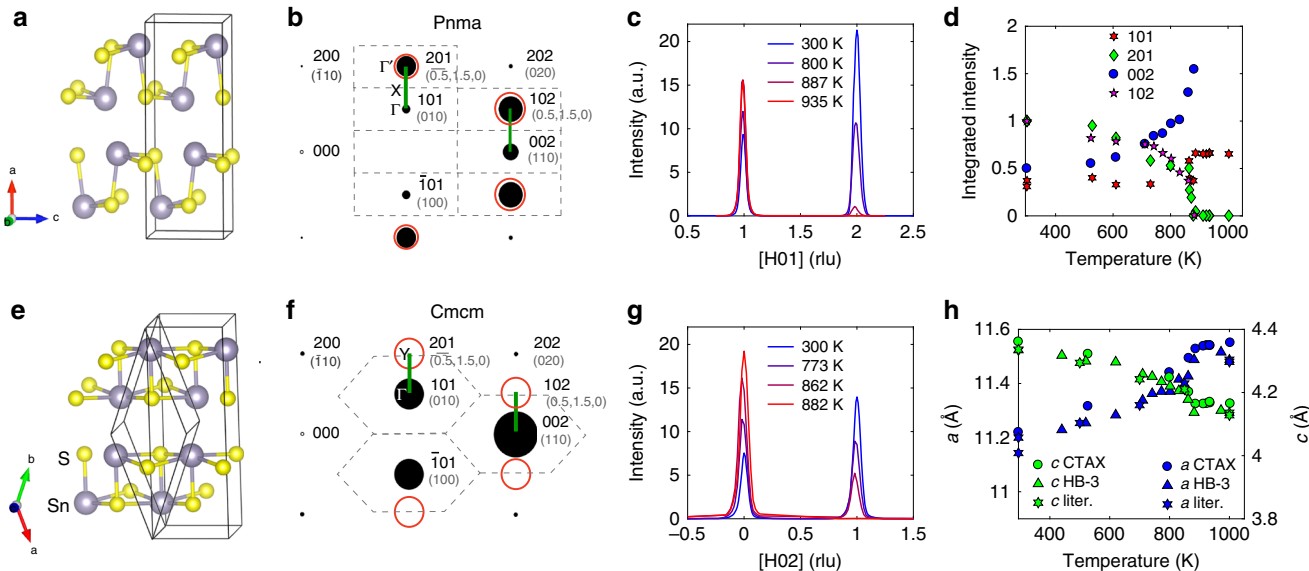

**Fig. 1 Crystal structure and phase transition. a, e** Crystal structures for the Pnma and Cmcm phases of SnS. **b, f** Neutron-weighted reciprocal space maps for the Pnma (300 K)[57] and Cmcm (1000 K)[11] phases in the (H0L) plane (conventional Pnma cell). Bold Miller indices are in Pnma notation, while those enclosed in parentheses are in primitive Cmcm notation. Brillouin zones for the conventional Pnma and primitive Cmcm cells are indicated by the dashed gray outlines, together with X and Y points, in (**b**) and (**f**), respectively. Superstructure Bragg peaks disappearing in the Cmcm phase are circled in red. Green segments correspond to directions measured with TAS. Single-crystal diffraction data on SnS for [H01] from CTAX (**c**) and for [H02] from HB-3 (**g**). **d** Evolution of SnS integrated peak intensities (±0.25 H (rlu)) and **h** lattice parameters (literature data from ref. [11]) with temperature. In **e** the rhombohedron shows the primitive cell, while the larger rectangular parallelepiped is the conventional cell. The coordinate system shown in (**e**) is for the primitive cell.

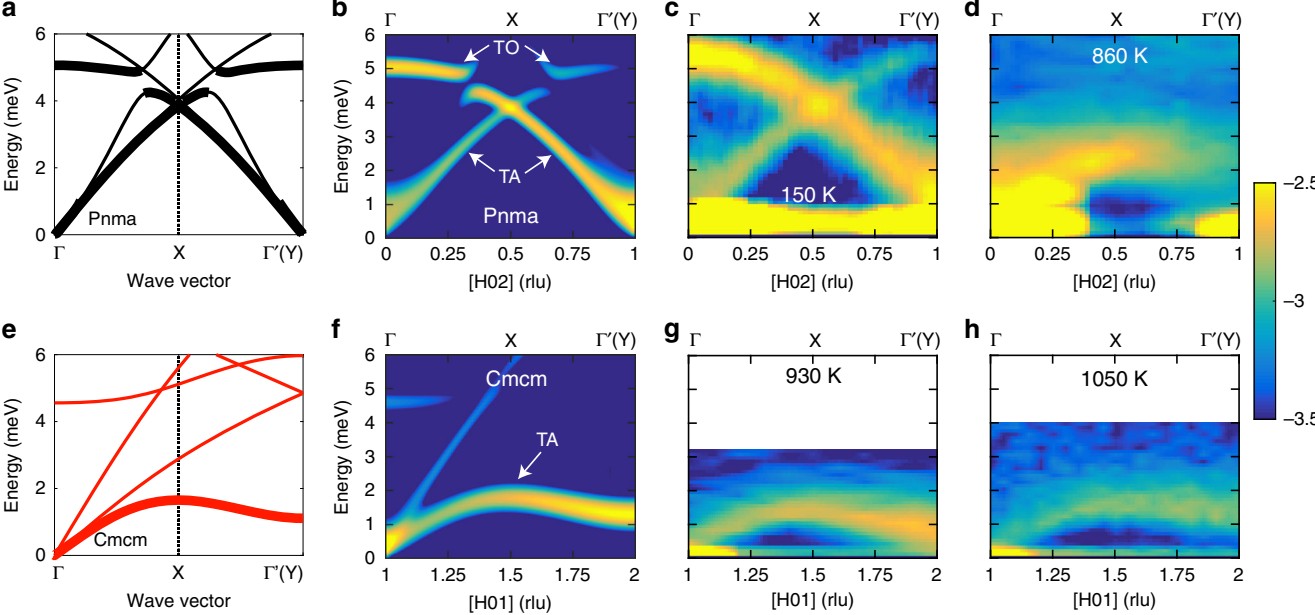

**Fig. 2 Extended anharmonic collapse of phonon dispersions.** Evolution of SnS phonon dispersions and dynamical susceptibility ($\chi''(\mathbf{Q}, E)$) across the structural phase transition. Upper panels are for the Pnma phase whereas lower panels show the Cmcm phase. Low energy dispersions computed in the Pnma phase with the harmonic approximation (**a**) and the Cmcm phase with TDEP at 800 K (**e**), for wave-vectors along [100] (Pnma notation). Thicker lines indicate which dispersions are seen in the experimental data. The same reciprocal path is plotted in both panels, corresponding to $\Gamma$–Y in the Cmcm phase and $\Gamma$–X–$\Gamma'$ in the Pnma phase, where $\Gamma'$ is a superlattice zone center (Fig. 1). High symmetry points of the Pnma Brillouin zone (BZ) are shown without parentheses whereas those for the Cmcm BZ are shown in parentheses. **b, f** Show the corresponding computed $\chi''(\mathbf{Q}, E)$ along [H02] and [H01] where Pnma notation for the Miller indices are used throughout. **c, d, g, h** Highlight the temperature evolution of $\chi''(\mathbf{Q}, E)$ measured with INS along the same direction in the irreducible BZ as (**b, f**). As indicated in (**b, f**), these reciprocal-space paths highlight low-energy $TA_c$ and $TO_c$ phonons polarized along c in the Pnma phase (**c, d**) and Cmcm phase (**g, h**). The paths shown in (**b–d**) are not exactly equivalent to those shown in (**f–h**) but show the same direction in the irreducible BZ and have very similar polarization conditions such that all show c-polarized modes. Data were collected on CNCS (**c**), HB-3 (**d**) and CTAX (**g, h**). Intensity is plotted in a $\log_{10}$ scale on color maps.

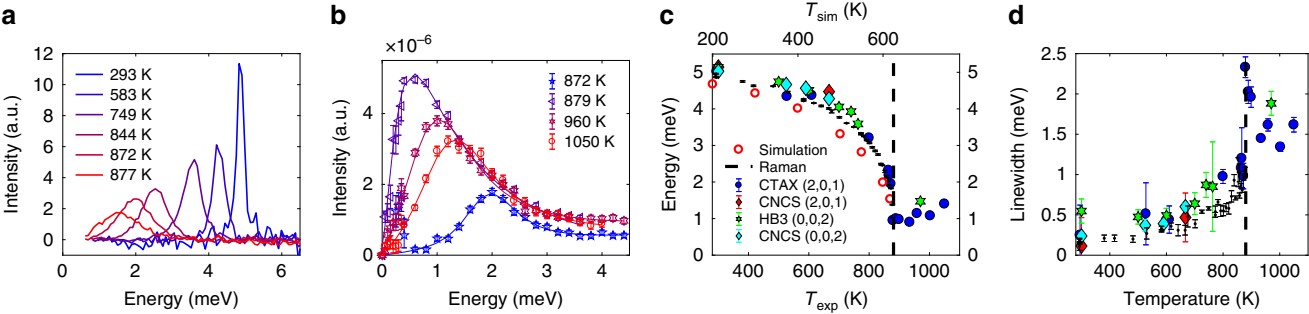

**Fig. 3 Soft mode behavior.** Temperature evolution of the soft phonon mode across the phase transition in SnS. **a** $\chi''(E)$ spectra from Raman scattering showing $TO_c$ mode as a function of temperature. **b** $\chi''(\mathbf{Q} = (2, 0, 1), E)$ phonon spectra from INS (CTAX) at temperatures near and above $T_C$. Error bars are from Poisson statistics and represent one standard deviation. **c**, **d** Soft mode energies and linewidths (full width at half maximum) extracted from fitting INS and Raman data (fitting procedure described in "Methods"), compared with simulations. The vertical dashed line represents $T_C$. The energies from simulations are derived from a linear interpolation of $\Phi^{(2)}$ between 0 K DFT and 600 K TDEP. In **c** the simulation temperatures are shown on a separate axis with the theoretical $T_C = 624$ K aligned with the experimental $T_C = 880$ K. Error bars for the neutron data in (**c**, **d**) represent one standard deviation obtained from fitting using Reslib whereas those for Raman data represent a 95% confidence interval from the fitting procedure.

high flux isotope reactor (HFIR) (details in "Methods"). Additional measurements with thermal neutrons were also performed with a thermal triple-axis spectrometer (HB-3) at HFIR and the wide angular-range chopper spectrometer (ARCS) at SNS. The temperature evolution of INS phonon measurements in SnS is shown and compared with simulations in Fig. 2. Panels a–d show the Pnma phase and e–h the Cmcm phase. Calculated phonon dispersions for the two phases are shown in panels a and e with polarization effects being clearly evident in b and f where the simulated $\chi''(\mathbf{Q},E)$ gives intensity only for the $c$-polarized $TA_c$ and $TO_c$ phonons. Experimental $\chi''(\mathbf{Q},E)$ maps for these dispersions are seen in the remaining panels showing the $TA_c$ and $TO_c$ modes, for phonon wavevectors along [100]. As seen in panels c and d, a drastic and extended softening (decreasing energy) of both branches occurs on warming in the Pnma phase. Above $T_C$, the re-emergence of a single $TA_c$ branch is observed in the Cmcm phase, as seen in panels g, h. Our measurements do directly reveal the condensation of a soft phonon mode at the transition (at $H =$ 2 in panel h for Cmcm and $H = 1$ in panel d for Pnma which are both superstructure peaks), but also show how the instability leads to an extreme softening affecting extended portions of TA branches in both phases, as well as the folded-back lowest-energy TO branch in the distorted Pnma phase. In the Cmcm phase, it is a zone boundary $TA_c$ mode that condenses at $\mathbf{Q}$'s corresponding to the Pnma super-structure Bragg peaks (e.g., (102) and (201)). The soft-mode re-emerges in Pnma as the back-folded zone-center (Γ) $TO_c$ mode. This striking evolution of the phonons was observed in multiple Brillouin zones and on measurements from different instruments, as shown in Supplementary Fig. 6. An almost identical behavior is observed in SnSe as seen in Supplementary Figs. 1, 2 but with energies shifted slightly lower relative to SnS due to the mass of selenium being higher than sulfur.

The extremely soft nature of the $TA_c$ branch in the SnS Cmcm phase near $T_C$ can be appreciated by considering data at 930 K (Fig. 2g). The branch rises from 0 meV at (1,0,1) and peaks at only 1.36 meV at (1.5,0,1) before decreasing to 0.90 meV at (2,0,1). This softening is extended across the entire dispersion as opposed to being localized at zone centers or boundaries. As $T$ is increased above $T_C$, the $TA_c$ branch stiffens back up along its entirety (Supplementary Fig. 6q–t). The extended nature of this ultra-low energy TA branch is captured well by the temperature dependent effective potential (TDEP) method[26] (see "Methods") for the Cmcm calculation (panels e, f). The TDEP method directly incorporates anharmonic renormalization by extracting

effective force constants through fitting the potential energy surface of temperature-dependent ab initio molecular dynamics (AIMD). The renormalized calculations show the continuous collapse of the $TO_c$ and $TA_c$ modes with temperature in the Pnma phase in Supplementary Fig. 4. The $TO_c$ softening occurs over a large temperature range, starting far below $T_C$. At 150 K the $TO_c$ mode at Γ has an energy of 5.3 meV and it is softened by 25.5% (to 3.9 meV) at 740 K, still 140 K below the transition.

We note that Fig. 2 panels c and d show systematic temperature variations in the intensities of phonon branches. This evolution of $\chi''(\mathbf{Q},E)$ reflects thermal changes in both Bragg peak intensities and anharmonic phonon eigenvectors (Supplementary Figs. 4, 5). At 150 K the $TA_c$ branch (<4 meV) shows the greatest intensity closest to (102), whereas at 860 K intensity is highest near (002). In Supplementary Fig. 6 we also observe drastic changes in the relative intensities of the $TO_c$ mode at (103) and (203) (panels a–e) and (002) and (102) (panels f–j). The thermal evolution of the $TO_c$ mode eigenvector obtained from our anharmonic phonon simulations is shown in Supplementary Fig. 27.

**Soft mode behavior.** We now proceed to quantify the energy, $E_0$, and linewidth ($\Gamma_{LW}$, full width at half maximum) of the soft mode. The phonon linewidths are inversely proportional to phonon lifetimes ($\tau$) while frequencies impact $\kappa_{lat}$ by altering group velocities ($v_g = |\nabla_k \omega|$), Bose–Einstein occupation factor ($n$ ($E$, $T$)) and the phase space for scattering. Extracting linewidths and energies from INS requires careful consideration of the instrument-specific resolution function, $R(\mathbf{Q}, E)$, which we calculated for each instrument, $\mathbf{Q}$-point and energy, and then convolved with fitting functions (details in "Methods"). In addition to INS data, we also collected high-resolution Raman spectra on single crystals of SnS and SnSe to probe zone-center optical modes (see Supplementary Note 4 for details), including the soft-mode in the Pnma phase. Phonon lineshapes were successfully modeled as damped harmonic oscillators (DHO), see Supplementary equation 5, which approximates to a Lorentzian line-shape in the usual case where the phonon energy $E_0$ is much larger than the damping $\Gamma_{LW}$. A strong departure from Lorentzian behavior is observed in the $\chi''(E)$ spectra (Fig. 3a, b) close to $T_C$, where the phonons are overdamped with $\Gamma_{LW} > E_0$. In this regime, the traditional phonon picture breaks down as modes no longer have well-defined oscillations.

The soft mode behavior is clearly observed in the Raman spectra in Fig. 3a where the $TO_c$ ($A_g$) mode softens and broadens upon approaching $T_C$ from below. The soft-mode disappears above $T_C$ in the Raman spectra, since it re-emerges as a zone-boundary $Y$ mode in the Cmcm phase. The lowest energy zone center optical mode in the Cmcm phase (seen in Fig. 2e, f at $\approx$ 4.5 meV) is Raman inactive from symmetry considerations[27] and as such we did not observe any peaks above $T_C$. Fits to INS data are shown in Fig. 3b, revealing the pronounced overdamping around $T_C$. In Supplementary Fig. 21, the DHO profiles extracted from fitting the INS data show the intrinsic lineshapes, corrected for instrumental resolution. Figure. 3c shows the temperature evolution of $E(TO_c)$, which is clearly continuous up to the condensation at $T_C$, and can be fitted with a power law, consistent with a second-order transition. Remarkably good agreement is found between INS measurements on different instruments and the Raman data. The reason why $E_0$ does not exactly reach zero is because of strong broadening in the DHO profile at $T_C$, a common feature of soft-mode transitions[28]. The soft mode behavior can be seen on both sides of $T_C$ in the INS data (CTAX), since no modes are symmetry-forbidden in INS, unlike Raman. In both phases the mode ($TO_c$ for Pnma and zone-boundary $TA_c$ for Cmcm) softens upon approaching $T_C$. This behavior is characteristic of soft mode transitions in general, as detailed in seminal studies[29].

The $TO_c$ linewidth shows a drastic increase at the transition (Fig. 3d), reaching a value 9.4$\times$ higher than at 295 K, corresponding to a drastically increased scattering rate. Importantly, this increase is far larger than the linear temperature dependence expected within low-order perturbation theory ($n(E, T) \propto T$ at high $T$). We observe a similar behavior in SnSe INS and Raman spectra (Supplementary Figs. 1–3). This divergence behavior near a second order (continuous) phase transition can be ascribed to critical behavior, similar to the divergence in the heat capacity (Supplementary Fig. 18). Further, a strong polarization dependence of the softening is revealed from the fitting of Raman data (Supplementary Fig. 20). Thus, the anharmonicity depends anisotropically on atomic motions through the bonding directionality. $A_g^1$ is $a$-polarized (out-of-plane) and exhibits significantly less softening than the $A_g^0$ and $B_{3g}$ modes polarized in the $b$–$c$ plane. This agrees well with the findings of Liu et al.[22] except here we focus on the lowest energy modes in the SnSe/SnS compounds whereas Liu et al.[22] only reported modes above 50 cm$^{-1}$.

We find a clear departure from mean field theory (MFT) over a large temperature range in both SnS and SnSe. MFT predicts $E(T) = A * |T - T_C|^\alpha$, where $A$ is a constant and $\alpha = \frac{1}{2}$ based upon an expansion of the free energy[28,30]. However, fitting the CTAX and Raman SnS (SnSe) data for the soft mode gives $\alpha = 0.23$ and 0.24 (0.23), respectively. Furthermore, the $c$ position of Sn atoms in the unit cell (offset from Cmcm mirror plane) can be identified as the order parameter of the Pnma–Cmcm transition. Fitting this order parameter (data taken from ref. [13]) gives $\alpha = 0.24$, in remarkable agreement with the soft phonon frequency. This exponent could potentially reflect the rather 2D nature of bonding, as well as long-range interactions from resonant bonding in SnS and SnSe (refs. [31,32]).

We now compare the experimental $TO_c$ evolution with simulations combining AIMD with TDEP[26] to incorporate anharmonic renormalization. As a baseline, phonon dispersions of the Pnma phase at 0 K were simulated within the harmonic approximation (Fig. 2a). Further, AIMD was performed at 600 K and combined with TDEP to extract renormalized second-order force-constants ($\Phi^{(2)}$) and phonon dispersions. A linear interpolation between the harmonic and renormalized $\Phi^{(2)}$ was

subsequently used to approximate the temperature dependence of $\Phi^{(2)}$ in the Pnma phase between 0 K and 600 K and extrapolated to the phase transition. The resulting temperature dependence of the $TO_c$ mode could be fitted with the same power law as above, resulting in a transition temperature $T_C = 624$ K and $\alpha = 0.28$, the latter of which is in reasonable agreement with experimental results. The underestimation of $T_C$ is consistent with recent anharmonic simulations on SnSe (ref. [18]). Simulation temperatures were scaled to match the experimental $T_C = 880$ K, therefore 600 K is scaled to 846 K when comparing with experiments. The resultant soft mode energies are plotted in Fig. 3c.

**Anharmonic renormalization and thermal conductivity.** Dramatic shifts in frequency and linewidth are observed for the soft mode. However, this giant anharmonicity, which is reflected in the changing $\Phi^{(2)}$ in Fig. 4d, is not limited to the soft mode. Supplementary Figs. 22–25 also show considerable phonon energy shifts and broadenings for different branches ($s$) and wavevectors (**q**). Extracting phonon energies allows for the calculation of group velocities ($v_g$), which is a crucial factor in the lattice thermal conductivity:

$$\boldsymbol{\kappa}_{lat} = \sum_{q,s} C_{v,q,s} \, \boldsymbol{v}_{g,q,s} \otimes \boldsymbol{v}_{g,q,s} \, \tau_{q,s}, \qquad (1)$$

where $C_v$ is the specific heat at constant volume. We extracted $v_g$ for the $TA_c$ branch at **Q** = (1.5,0,1), (1.8,0,1) from the CTAX data over a large temperature range, as shown in Fig. 4a. From 295 to 864 K, a drastic reduction of 35% and 77% is observed in $v_g$ at **q** = (0.2, 0, 0) and (0.5, 0, 0), respectively. By contrast, $v_g$ for the $LO_c$ mode along [001] increases for small **q** as shown in Fig. 4b. We note that this complex and anisotropic $v_g$ behavior is qualitatively reproduced in the simulations. However, based on our simulations, $\bar{v}_g$ averaged over all branches and throughout the Brillouin zone remains almost constant: $\bar{v}_g = 948$ ms$^{-1}$ at 0 K and 973 ms$^{-1}$ at 600 K. This anisotropic temperature dependence of the group velocities requires carefully investigating the effects of renormalization on $\kappa_{lat}$, which we now discuss.

Phonon renormalization with temperature is currently neglected in the majority of first-principles thermal conductivity simulations. In a widely embraced approach pioneered by Broido et al.[33], the phonon Boltzmann transport equation (BTE) is solved using first-principles simulations of second and third-order force-constants, $\Phi^{(2)}$ and $\Phi^{(3)}$ based on low-order perturbation theory[33–36]. The scattering rates ($\Gamma$) in the BTE are usually obtained from three-phonon scattering processes (sometimes extended to higher order), but energies and group velocities are typically from the harmonic dispersions. While this formalism has been successful in analyzing many materials and has led to important predictions that were later verified, such as ultra-high thermal conductivity in boron arsenide[37–39], the range of anharmonic strength over which it remains valid is poorly established.

Here, we show that anharmonic phonon renormalization, achieved through using the TDEP method combined with temperature-dependent AIMD, can have large effects on thermal conductivity. The anharmonic renormalization in SnS leads to a strong reduction in the calculated $\kappa_{lat}$ of 39% just by changing $\Phi^{(2)}$ (0 K harmonic $\rightarrow$ AIMD-TDEP at 600 K), whilst keeping $\Phi^{(3)}$ unchanged. Group velocities directly depend on the phonon frequencies, which can be obtained from either the harmonic or renormalized phonon models. In addition, scattering rates $\Gamma_s$ for different phonon branches $s$ also depend on energies (Eq. (2))[40], although more indirectly, through the scattering phase space and

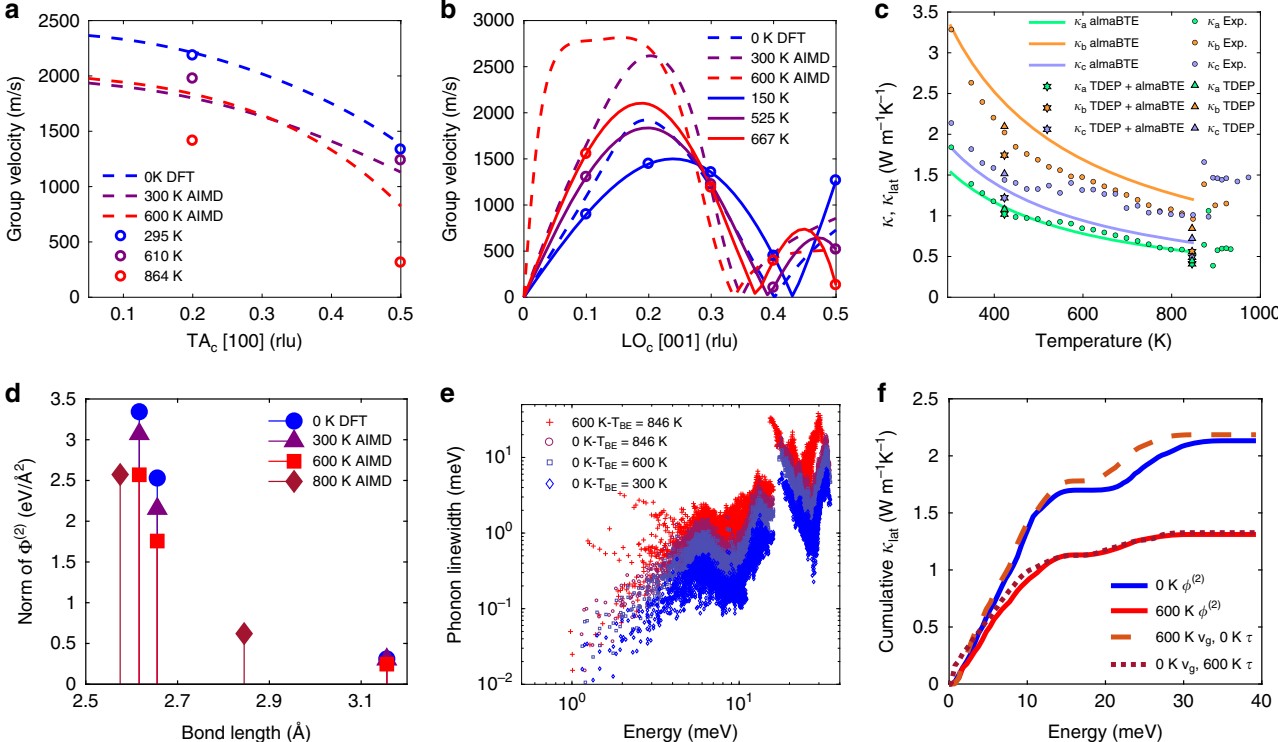

**Fig. 4 Phonon renormalization and effect on thermal transport. a** Group velocities extracted from experiments and simulations (see "Methods" for details) for the $TA_c$ along [100] (measured with CTAX at $\mathbf{Q} = (1.5, 0, 1)$ and $(1.8,0,1)$) and **b** $LO_c$ along [001]. Open circles show $\mathbf{Q}$ corresponding to experimental measurements, solid lines are for $v_g$ from experiments and dashed lines for $v_g$ from simulations. **c** Measured $\kappa$ (circles) and calculated $\kappa_{lat}$ (lines and markers) along the $a$, $b$, and $c$ crystallographic directions. Lines represent $\kappa_{lat}$ computed with almaBTE and triangles are values from TDEP, while stars are from almaBTE, but with renormalized $\Phi^{(2)}$ from TDEP. **d** Norm of $\Phi^{(2)}$ from DFT (0 K) and TDEP (Pnma at 300, 600 K and Cmcm at 800 K). **e** Phonon linewidths calculated with $\Phi^{(2)}$ from 0 K DFT ($T_{BE} = 300$, 600 and 846 K) and 600 K TDEP ($T_{BE} = 846$ K) with fixed $\Phi^{(3)}$ (almaBTE). **f** Cumulative $\kappa_{lat}$ ($T_{BE} = 300$ K) from RTA with $\Phi^{(2)}$ from 0 K DFT ($v_{g,0K} + \tau_{0K}$) and 600 K TDEP ($v_{g,600K} + \tau_{600K}$), as well as mixed group velocities and lifetimes ($v_{g,600K} + \tau_{0K}$, and $v_{g,0K} + \tau_{600K}$).

**Table 1 Lattice thermal conductivity ($\kappa_{lat}$, Wm$^{-1}$K$^{-1}$) from BTE with different $\Phi^{(2)}$ and Bose–Einstein temperatures ($T_{BE}$) for thermal occupation factor. $\kappa_{lat,i}$ is along direction $i$ while $\overline{\kappa_{lat}}$ is the orientation averaged value.**

| $T_{BE}$ | $\Phi^{(2)}$ | $\kappa_{lat,a}$ | $\kappa_{lat,b}$ | $\kappa_{lat,c}$ | $\overline{\kappa_{lat}}$ | $\kappa_{exp}$ |
|---|---|---|---|---|---|---|
| 300 K | DFT | 1.54 | 3.35 | 1.85 | 2.25 | 2.42 |
| 300 K | QHA-DFT | 0.59 | 1.80 | 1.52 | 1.30 | – |
| 600 K | AIMD 600 K | 1.15 | 1.58 | 1.37 | 1.37 | 0.84* |
| 846 K | AIMD 600 K | 0.41 | 0.56 | 0.49 | 0.49 | 0.84* |

*Experimental $\kappa$ at 848 K.

the Bose–Einstein occupation factor (Eq. (3)).

$$\Gamma_s = \frac{\hbar\pi}{16} \sum_{s_1 s_2} \left| \Phi_{s,s_1,s_2} \right|^2 \left[ (n_{s_1} + n_{s_2} + 1) \, \delta(\omega_s - \omega_{s_1} - \omega_{s_2}) \right.$$
$$\left. + 2(n_{s_1} - n_{s_2}) \, \delta(\omega_s - \omega_{s_1} + \omega_{s_2}) \right] \quad (2)$$

$$n(\omega_{q,s}, T_{BE}) = \left( \exp(\hbar\omega_{q,s}/k_B T_{BE}) - 1 \right)^{-1} \quad (3)$$

The lattice thermal conductivity was calculated with almaBTE[41] as well as TDEP (see "Methods"). Our $\kappa_{lat}$ calculations agree well with our direction-dependent measurements of $\kappa_{tot}$ on SnS crystals (details in "Methods"), shown in Fig. 4c. We note that the $\kappa_{lat}$ obtained with TDEP agrees better with experimental values than the $\kappa_{lat}$ obtained with almaBTE, reflecting the

importance of renormalization effects. Our measured data are in good agreement with previous single-crystal measurements of SnS[42], in particular showing a similar anisotropy: $\kappa_b > \kappa_c > \kappa_a$ (Supplementary Fig. 16). The electronic component $\kappa_{el} = \kappa_{tot} - \kappa_{lat}$ is expected to only contribute a few percent of the total[42]. Our calculations also agree with trends in previous simulations[35,36], yielding $\kappa_{b,lat} > \kappa_{c,lat} > \kappa_{a,lat}$, but the absolute values differ significantly (Supplementary Fig. 15). We first consider the effect of unit cell expansion and shape at the quasiharmonic (QHA) level. Comparing the first two lines in Table 1, we see that $\kappa_{lat}$ at occupation temperature $T_{BE} = 300$ K is strongly suppressed when using $\Phi^{(2)}$ calculated with an expanded unit cell in QHA (see "Methods") compared to results from $\Phi^{(2)}$ evaluated on the theoretical relaxed cell. In particular, the expanded cell, representative of $T = 862$ K, results in 62% suppression along $a$. Further, the anisotropy between $\kappa_{b,lat}$ and $\kappa_{c,lat}$ decreases, reflecting the nearly tetragonal cell shape near $T_C$ ($b = 4.07$ Å, $c = 4.05$ Å[24]) and capturing some of the experimental trend.

To investigate intrinsic renormalization effects, $\Phi^{(3)}$ were combined with $\Phi^{(2)}$ from either harmonic 0 K or 600 K TDEP calculations. These $\Phi^{(2)}$ are then used to calculate $\kappa_{lat}$ (almaBTE) at different $T_{BE}$ with fixed $\Phi^{(3)}$. The strong reduction in $\kappa_{lat}$ upon changing $\Phi^{(2)}$ is found to be dominated by changes in phonon lifetimes rather than group velocities, reflecting changes in the phase space from the pronounced phonon renormalization. To establish this, we calculate the change in $\kappa_{lat}$ from changing $\Phi^{(2)}$ calculated at 0 K (harmonic) to those obtained at 600 K (AIMD-TDEP), whilst fixing $T_{BE} = 300$ K (Table 1, first versus third row and Fig. 4f blue curve versus red curve). This drastically reduces

the direction-averaged $\bar{\kappa}_{lat}$ from 2.25 to 1.37 W m$^{-1}$ K$^{-1}$. This reduction of 39% in $\bar{\kappa}_{lat}$ results purely from phonon renormalization effects as $\Phi^{(3)}$ were kept fixed. Further, we separate the effects of renormalization on $v_g$ and $\tau$ in Eq. (1), which both depend on $\Phi^{(2)}$. A change from $v_{g,0K}$ to $v_{g,600K}$, whilst keeping $\tau_{0K}$ fixed, has a relatively small effect on thermal conductivity, as seen in the cumulative $\kappa_{lat}$ plot in Fig. 4f. In fact, this actually acts to slightly increase $\kappa_{lat}$ by 2.3%, which arises from a mixture of increasing and decreasing $v_g$ along different branches upon warming, as seen in Fig. 4a, b. Conversely, a change from $\tau_{0K}$ to $\tau_{600K}$, while fixing $v_g = v_{g,0K}$, leads to a large reduction of 38% in $\kappa_{lat}$ (dotted brown curve), which is almost equivalent to the 39% reduction found when using $\tau_{600K}$ with $v_{g,600K}$ (red curve). Thus, the suppression in $\kappa_{lat}$ from $\Phi^{(2)}$ renormalization occurs mainly through an increase in scattering rates (linewidths), shown in Fig. 4e, especially pronounced for low-energy TO and TA modes below 5 meV. In Supplementary Table 1, we benchmark the calculated linewidths against those obtained from fitting Raman spectra and find good agreement. The linewidths increase across the entire spectrum when thermally renormalizing $\Phi^{(2)}$ while keeping $\Phi^{(3)}$ fixed (purple circles versus red crosses), illustrating the importance of anharmonic renormalization of dispersions.

Using TDEP, we computed $\kappa_{lat}$ with renormalized force-constants ($\Phi^{(2)}$ and $\Phi^{(3)}$) from AIMD at 300 K and 600 K, with fixed $T_{BE} = 300$ K to isolate renormalization effects. Renormalizing both $\Phi^{(2)}$ and $\Phi^{(3)}$ suppresses $\kappa_{lat}$ from 2.10 W m$^{-1}$ K$^{-1}$ to 1.77 W m$^{-1}$ K$^{-1}$, (a 16% decrease). Including only the effect of renormalized group velocities would actually increase $\kappa_{lat}$ from 2.10 W m$^{-1}$ K$^{-1}$ to 2.24 W m$^{-1}$ K$^{-1}$, owing to the increased average group velocity (from 1041 m s$^{-1}$ to 1117 m s$^{-1}$). On the other hand, changing the phonon lifetimes from 300 K TDEP to 600 K TDEP results in a thermal conductivity suppression from 2.10 W m$^{-1}$ K$^{-1}$ to 1.59 W m$^{-1}$ K$^{-1}$, a 24% suppression. Thus, our TDEP simulations further support our conclusion that the change in scattering rates is the dominant renormalization effect, compared to the more minor effect of group velocity renormalization.

**Conclusion.** Our results reveal drastically temperature-dependent and anisotropic lattice dynamics of SnS and SnSe. Spectacular softening is observed for extended regions of the TA$_c$ and TO$_c$ dispersions along [100] in Pnma below $T_C$, and an extremely soft TA$_c$ branch persists in the Cmcm phase. However, renormalization effects were not limited to these dispersions and instead were found to be significant throughout the Brillouin zone leading to strong renormalization and pronounced anisotropic changes in group velocities. While not included in typical calculations based on low-order perturbation theory, similarly strong anharmonic renormalization effects are expected to occur in many materials near structural transitions, including halide perovskites, oxide ferroelectrics, or thermoelectrics near instabilities (e.g., PbTe, PbSe, GeTe, Cu$_{2-x}$Se) besides SnS and SnSe. This work provides impetus for future first-principles $\kappa$ simulations to consider renormalization effects and indeed to re-examine older work within this context. Furthermore, our results provide additional motivation to investigate materials exhibiting lattice instabilities in the search for ultra-low $\kappa$ materials.

## Methods

**Sample synthesis.** Single crystals of SnS were synthesized using a modified Bridgman technique with high purity Sn and S (Alfa Aesar, 99.999%). A poly-crystalline material was first synthesized by slowly heating the elements in a fused silica ampoule to 500 °C for 10 h, followed by 10 h at 950 °C. The growth of large crystals occurred in a tapered, fused silica ampoule while cooling to ~830 °C using a rate of 0.4–0.5 °C h$^{-1}$. This was followed by a 24 h anneal before cooling to 500 °C and shutting off the furnace. The quality of the crystals was checked with X-ray and neutron diffraction. Single crystals of SnSe were synthesized as described in ref. [12]. Typical crystals used in neutron scattering experiments had a mass of about 10 g.

To prevent sublimation and oxidation during high-temperature measurements of SnS and SnSe on CTAX and HB-3, the samples were encapsulated in a double-walled quartz ampoule. Prior to sealing, the ampoule was flushed three times with argon gas, and then a rough vacuum was pulled on the sample space. The samples were held in place inside their ampoule with quartz slides and quartz wool, with alignment markings visible on the sample itself. The quartz ampoules containing the samples were attached to the furnace mounting stick using high-temperature FeCrAl thermocouple wire.

**Inelastic neutron scattering measurements.** Triple-axis (TAS) measurements were performed on single crystals of SnS using the cold neutron triple-axis spectrometer (CTAX) and with thermal neutrons on the HB-3 triple-axis spectrometer at the high flux isotope reactor (HFIR) at Oak Ridge National Laboratory (ORNL). Both experiments were conducted with the [010] axis vertical such that the [H0L] scattering plane was accessible. The HB-3 measurements were made with a final energy of $E_f = 14.7$ meV and horizontal collimation settings of 48′-20′-20′-70′ ($\Delta E \approx 0.7$ meV). PG002 monochromators and analyzers were used. The HB-3 experiment was focused on [H02] direction for $0 \leq H \leq 1$ whereas CTAX investigated the [H01] direction for $1 \leq H \leq 2$ owing to limitations of the accessible wave-vector transfer, **Q**. The CTAX measurements were undertaken in fixed initial energy mode ($E_i = 5$ meV). Horizontal collimation settings of 30′-100′-80″-120′ ($\Delta E \approx 0.3$ meV) were utilized. For both sets of triple-axis measurements an ILL niobium foil vacuum furnace was used with a thermocouple mounted close to the sample position. TAS measurements are performed at a specific point in momentum space but many TAS scans along a direction in **Q**-space can be combined to generate a dispersion.

The cold neutron chopper spectrometer (CNCS)[43] at the spallation neutron source (SNS) was used to make time-of-flight (TOF) measurements on SnS single-crystals. An incident energy of 17 meV was used and the following temperatures were sampled: 150, 300, 525, 590, and 667 K (temperature at the sample position). A boron nitride mask and radial collimator were used to block extra scattering from the sample mount and an aluminum foil heat shield was used to minimize temperature gradients. At each temperature the sample was rotated in one degree steps over a large angular range (50° for $T = 150$, 590 K, 91° for $T = 300$, 667 K and 60° for $T = 525$ K) in order to collect data for multiple crystal orientations. Resultant data were reduced using Mantid algorithms to recreate the four-dimensional dynamical structure factor, $S(\mathbf{Q}, E)$[44]. The Horace software package was then used to 'slice' the data along specified **Q**-directions to probe the phonon dispersions[45]. The data were typically integrated over $\pm 0.05$ or 0.10 rlu in the **Q**-directions perpendicular to the dispersion. Similarly, one-dimensional cuts were created at specific **Q** by integrating in all three directions around the point of interest.

Phonon linewidths and energies were extracted from TAS spectra by convolving fit functions with the instrumental resolution as implemented in the ResLib software package[46]. The resolution function was calculated using the Popovici approximation[47]. For TOF measurements Monte-Carlo ray tracing simulations were used to approximate the instrumental resolution using the MCViNE program[48]. The accuracy of this approach was tested by comparing the computed resolution function to the width of the incoherent elastic peak. For each peak the resolution function was simulated and integrated over all Q. The resultant R(E) was then fitted with a gaussian. Similarly, as for the TAS measurements the convolution of this gaussian and a damped harmonic oscillator (DHO), representing the intrinsic phonon, could then be fitted to match the experimental data in order to extract the phonon linewidths. DHO signals were first multiplied by the occupation factor before convolution with $R(\mathbf{Q}, E)$ such that $S(\mathbf{Q}, E)$ spectra could be fit directly. This workflow is demonstrated in Supplementary Fig. 29. Fitting functions contained several components representing phonons, Bragg peaks and incoherent elastic signals.

**Raman spectroscopy.** Raman spectra were collected on a Horiba T64000 Raman spectrometer in the double subtractive mode with an Olympus SLMPLN 50X Achromat microscope objective. A 532 nm Nd:YAG laser (~865 µW at sample position) was used to excite the sample. Each spectrum was an accumulation of 10–15 measurements with a 120 s exposure time. Single crystals of SnS were measured as a function of temperature (ambient to 877 K) using a Linkam TS1500 temperature controlled stage. The sample was positioned onto a sapphire disk within the Linkam stage which was then sealed and the measurements were performed under vacuum. Due to the layered nature of the materials the sample cleaves into flakes with the $a$ (long) axis (Pnma notation) perpendicular to the face of the flake. Measurements were performed with the laser beam parallel to the $a$ axis. At each temperature a spectrum was collected from the sapphire disk and this was used later to calibrate the temperature. In order to fit the peaks of collected spectra the background was first subtracted from the spectrum. Multiple Voigt functions were then used to fit the peaks and extract the centroid energies and linewidths. The Gaussian component of the Voigt represented the instrument resolution contribution whilst the Lorentzian component represented the phonon. The FWHM of the Gaussian (0.09 meV) was obtained by fitting the laser line in the

Horiba T64000 Raman spectrometer in the same configuration used for all measurements described within this paper.

**First-principles simulations**. First-principles calculations were performed based on density functional theory (DFT), as implemented in the Vienna Ab initio Simulation Package (VASP)[49–51]. The projector-augmented-wave (PAW) potential and local-density approximation (LDA) were used for all calculations[52,53], following our previous investigations of SnSe (refs. [12,25,32]). The S($3s^23p^4$) and Sn ($4d^{10}5s^25p^2$) electrons were treated as valence states. The SnS Pnma structure was fully relaxed starting with the experimental structure, with force convergence 0.1 meV/Å, plane-wave cutoff 500 eV and $6 \times 12 \times 12$ Γ centered K-points. The lattice constants after full relaxation were $a = 10.97$ Å, $b = 3.95$ Å and $c = 4.19$ Å, which agree reasonably well with experimental results ($a = 11.20$ Å, $b = 3.99$ Å, $c = 4.33$ Å at 296 K)[24] but are slightly smaller in line with noted behavior of the LDA approximation. The SnS Cmcm structure was fully relaxed as well, starting from the experimental Cmcm experimental structure and relaxed with the same settings as the Pnma structure, with force convergence 0.01 meV/Å. The Cmcm crystallographic axis were chosen to be consistent with the Pnma structure with ($a > c \geq b$). The lattice constants after full relaxation are $a = 11.16$ Å, $b = 4.00$ Å and $c = 4.00$ Å which are also consistent with experimental lattice constants at 883 K: $a = 11.49$ Å and $b = c = 4.16$ Å (ref. [24]).

Phonon dispersion calculations were performed according to several schemes. Harmonic phonon simulations were performed within the finite displacement method as implemented in Phonopy[54], with displacement 0.01 Å. We used $2 \times 4 \times 4$ supercells of the fully relaxed Pnma structure with a reduced Γ-centered $3 \times 5 \times 5$ K-point mesh and 500 eV cutoff energy. However, in the Cmcm structure, the phonon dispersion was found to be unstable in the harmonic approximation, in agreement with previous reports for SnSe in the Cmcm structure[15,17,32,55]. Quasi-harmonic approximation (QHA) calculations were performed by applying experimental thermal expansion from 296 K to 862 K (ref. [24]) to the relaxed lattice parameters, resulting in $a = 11.20$ Å, $b = 4.07$ Å and $c = 4.05$ Å at 862 K.

To further investigate the temperature dependence of phonons in both phases, ab initio molecular dynamics (AIMD) trajectories were calculated within a canonical ensemble (NVT) using a Nosé-Hoover thermostat. Γ only K-point and 500 eV cutoff energy were used for the $2 \times 4 \times 4$ supercell of Pnma conventional cell in AIMD calculations at 300, 400, 500, 600, and 700 K extracted with an interaction cutoff distance $r = 7.887$ Å in the TDEP method[26]. The same settings were used for a $2 \times 4 \times 4$ supercell of conventional Cmcm unit cell at 550, 600, 700, 800, and 900 K, with force constants extracted for $r = 6.945$ Å. Based on the forces sampled in AIMD, which includes anharmonic components of the potential, we used the TDEP method to compute renormalized effective force constants[26]. For our calculation, around a total number of 1.5 ps with 1 fs/step for each temperature were calculated, and initial 0.5 ps were discarded. 1 ps was used to extract renormalized second order force constants at each temperature.

The lattice thermal conductivity was calculated with almaBTE[41]. Third-order force constants were calculated with a $2 \times 4 \times 4$ supercell, Γ-only **k**-grid, 400 eV cutoff energy, and using 800 independent configurations. The total energy for each configuration was converged to $10^{-8}$ eV. The phonon **q**-point mesh for thermal conductivity calculation was $15 \times 15 \times 15$ (a comparison with using $11 \times 21 \times 21$ is shown in Supplementary Fig. 16b) with a third-order force constant cutoff radius of 6.12 Å. Convergence of $\kappa_{lat}$ was investigated with respect to **q**-point mesh and cutoff radius and the results are shown in Supplementary Fig. 13. Thermal conductivity and phonon linewidths are calculated using almaBTE[41] by solving the BTE equation either iteratively or using the relaxation-time approximation (RTA), with either the 0 K harmonic $\Phi^{(2)}$ or the renormalized $\Phi^{(2)}$ from TDEP at 300 and 600 K. Phonon linewidths, $\Gamma_s$, were calculated using the inverse $\tau$ from RTA, as $\tau_s^{-1} = \pi \Gamma_s$ (ref. [56]).

Third-order force constants were also extracted with the TDEP method from AIMD at 300 K and 600 K, using an interaction cutoff distance $r = 6.12$ Å. Second-order and third-order force constants from TDEP were used to calculate the lattice thermal conductivity, also with TDEP, with a phonon **q** mesh of $15 \times 15 \times 15$ points, which provided reasonable convergence, as shown in Supplementary Fig. 14. In TDEP, the thermal conductivities are calculated using the adaptive Gaussian integration method and solving the BTE equation iteratively at $T_{BE} = 423$ K and $T_{BE} = 846$ K.

**Thermal conductivity measurements**. Experimental orientation-dependent thermal conductivities for SnS were obtained from the heat capacity ($C_p$) measured with differential scanning calorimetry (DSC) and thermal diffusivities ($D$) measured using the transient light flash method on single crystals of SnS cut along the primary crystallographic directions. Thermal conductivity can be calculated as the product of density ($\rho$), heat capacity ($C_p$), and thermal diffusivity ($D$):

$$\kappa = \rho C_p D \tag{4}$$

Heat capacity measurements were performed on a small (66.1 mg) single crystal of SnS using a differential scanning calorimeter (DSC 404 F1 Pegasus, Netzsch). Measurements were performed under flowing Ar with a heating rate of 10 K/min. Three consecutive DSC runs were performed with empty crucible (baseline), sapphire (reference) and SnS (sample) in order to calculate the heat capacity

according to DIN 51007. Thermal diffusivity was measured using the light flash method with a Netzsch LFA 467HT apparatus. X-ray Laue was used to align a large single crystal ingot of SnS for cutting. Three different pieces (plate-like shapes) were then cut, each with a different principal crystallographic axis perpendicular to the plate face. The thickness of these pieces ranged from 1.25 to 2.00 mm and they were coated with graphite, to improve absorption and emissivity, prior to measurement. Three measurements were taken at each temperature in order to calculate the mean $D$ and data was also recorded on cooling to confirm reproducibility. Thermal diffusivity was calculated from the measured sample response according to the Cape-Lehman method. Sample density was taken to be 5.22 g cm$^{-3}$.

## Data availability

Source data are provided with this paper. Data that support the plots within this paper and other findings of this study are available from the corresponding authors upon reasonable request.

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

## Acknowledgements

We thank Olle Hellman for providing access to and support with the TDEP software package. We thank Jason Craig at ORNL for sample encapsulation, and Rebecca Mills at ORNL for help with neutron furnaces. We thank Chen Li for help with initial screening in SnS crystals. Neutron scattering data collection and analysis by TLA and OD was supported the U.S. Department of Energy, Office of Science, Basic Energy Sciences, Materials Sciences and Engineering Division, under Award No. DE-SC0019299. First-principles simulations (SY) and neutron data collection (JN, DB) were supported by the U.S. Department of Energy, Office of Science, Basic Energy Sciences, Materials Sciences and Engineering Division, under the Early Career Award No. DE-SC0016166. Sample synthesis (AFM) was supported by the U.S. Department of Energy, Office of Science, Basic Energy Sciences, Materials Sciences and Engineering Division. The use of Oak Ridge National Laboratory's Spallation Neutron Source and High Flux Isotope Reactor was sponsored by the Scientific User Facilities Division, Office of Basic Energy Sciences, U.S. DOE. Theoretical calculations were performed using resources of the National Energy Research Scientific Computing Center, a U. S. DOE Office of Science User Facility supported by the Office of Science of the U.S. Department of Energy under Contract No. DE-AC02-05CH11231. Initial Raman scattering measurements were conducted at the Center for Nanophase Materials Sciences, which is a DOE Office of Science User Facility.

## Author contributions

T.L.A. and S.Y. contributed equally to this work. Neutron scattering measurements and analysis were performed by T.L.A., J.L.N., D.B., and O.D. with support from G.E., D.P., S.C., and T.H. S.Y. performed the first-principles simulations. Sample synthesis was performed by A.F.M. T.L.A. performed the transport and Raman measurements. Initial Raman measurements on SnSe were performed by A.P. J.Y.Y.L. provided help with CNCS resolution analysis. The project was supervised by O.D.

## Competing interests

The authors declare no competing interests.
