## [Peer Review File · Nature Communications]

Editorial Note: Parts of this peer review file have been redacted as indicated to remove third-party material where no permission to publish could be obtained.

Reviewers' comments:

Reviewer #1 (Remarks to the Author):

Recommendation: minor revision

Comments:

Tin sulfide (SnS) is a type of layer structure material with strong anharmonicity, and aroused much attention for its promising energy conversion applications. This work reported the phase structure transition progress of SnS and further studied its phonon dispersions by experimental measurements and theoretical calculations. The investigation of atomic scale structure evolution uncovered the phonon scattering and soft-mode mechanism during phase transition. It is a comprehensive and solid work for understanding the phonon transports in materials with structural phase transition. This paper is well written, the experimental results and theoretical calculations are self-consistent, and the discussion is reasonable. Here are some comments and suggestions should be considered as below.

1. From inelastic neutron scattering (INS) experiments, the authors observed striking "softening" behaviors of TA and TO modes before phase transition (Pnma), however the TA branch stiffens up rather than softening with increasing temperature at Cmcm phase region. Therefore, what's the difference of this temperature-dependent behavior for TA mode in Pnma and Cmcm? And what effects will that have on the phonon propagation or thermal transport during phase transition?
2. I am a bit of confusion that what does the sharp increment of linewidths mean near the phase transition in Fig. 3d, does it relate to the abrupt change of thermal lattice conductivity corresponding to T_c ?
3. In the first-principles thermal conductivity simulations, how the authors estimate the contribution of acoustic and optical branches, and their temperature-dependent group velocities? In addition, the authors have calculated the temperature dependence of lattice conductivity before 800 K, how about the thermal conductivity near phase transition (from Pnma to Cmcm)?
4. In the discussion part of anharmonic renormalization, how the authors realize the phonon renormalization with temperature to simulate the thermal conductivity? How does this renormalization effect lattice conductivity?

Reviewer #2 (Remarks to the Author):

Review report for the manuscript titled, "Extended anharmonic collapse of phonon dispersions in SnS" by T. Lanigan-Atkins et al.

The authors have performed extensive INS and High resolution Raman measurements to demonstrate the phase transition mechanism in SnS from Pnma phase to Cmcm phase. The authors have measured the thermal conductivity of SnS and compared with first principles calculations using TDEP/AlmaBTE codes. The manuscript is very clearly written and figures are informative. I have a few questions/comments which need to be addressed before I recommend this manuscript for publication in Nat. Comm.

Major comments:

1. The authors claim that, in fig. 3a, the Raman peak disappears after the phase transition to Cmcm phase. But in fig. 2e, there seems to be a zone center optic phonon mode (at Gamma) in a similar

energy scale as in fig. 3a. So unless that optic mode in fig. 2e is not Raman active, I would expect that mode to show up in the Raman spectrum. Can the authors explain/comment on this point?

2. Can the authors give a physical reasoning as to why phonon renormalization from TDEP increases the scattering rates compared to the 0 K force constants (see fig. 4e)? As far as I know, the renormalization reduces the interactions among phonons (see Cowley's paper: R A Cowley 1968 Rep. Prog. Phys. 31 123 and Wallace's book: Thermodynamics of Crystals).

3. The authors have essentially used the comparison between two calculations (one with 0 K, harmonic Φ_2 and one with TDEP Φ_2 at 600 K, both with TDEP Φ_3) to conclude that the differences in the scattering rates rather than the group velocities (upon renormalization) has the dominant "renormalization effect" on the thermal conductivity. I don't think this is a fair comparison. Using 0 K Φ_2 and TDEP@600 K Φ_3 will lead to a completely different free energy manifold than the TDEP Φ_2 and Φ_3 at 600 K. In order to draw this conclusion, a fair comparison would be between 0 K Φ_2 and 0 K Φ_3 (like the supercell displacement method implemented in AlmaBTE) and TDEP Φ_2 and TDEP Φ_3 .

4. Does the free energy calculation in TDEP also predict that the Pnma phase is the stable phase below $\sim T_c$ and Cmcm phase is the stable phase above T_c ? Further, what is the theoretical T_c predicted by TDEP?

Minor comments:

1. Why did the authors decide to use AlmaBTE for the BE solution and not the BE solver in TDEP itself?

2. Did the authors solve the BE iteratively for the thermal conductivity? Is there a big difference between the Relaxation Time Approximation and the full solution of the BE for the thermal conductivity?

3. Please define DHO in the main text. It is only defined in the Methods section first.

Reviewer #3 (Remarks to the Author):

This is a very nice paper on lattice dynamics as a function of temperature of SnS and SnSe. The paper presents an impressive set of new inelastic neutron scattering phonon measurements, with a detailed study of the temperature dependence of the dispersion but also of the intensity distribution $S(Q,E)$ a crucial parameter when comparing with simulations.

A spectacular evolution of the $S(Q,E)$ response function is observed near the phase transition, with a phonon softening on a very large portion of the Brillouin zone for specific propagation vectors. Results are compared to state of the art atomic scale simulation using DFT but also energy renormalisation through AIMD and TDEP. The agreement between experiment and simulation is striking.

The effect of the phonon renormalisation on the lattice dynamics has also been studied in details : the strong reduction of the lattice thermal conductivity is mainly due to a strong decrease of phonon lifetime due to renormalisation, whereas the group velocity renormalisation does not play a major role.

The paper is of a broad interest, and brings in important results both in the field of soft mode phase transition and for the understanding of their low thermal conductivity. The paper certainly deserves to be published in Nature Communication.

However the authors should answer the following remarks and provide some changes in their manuscript.

1- The first remark concern the readability of the paper. The paper gathered an impressive set of both experimental and simulation results, as illustrated by the large supplementary information file.

However I have the feeling that reading from a non specialist is sometime a bit difficult.

I made the following suggestions :

- Structure and space groups, indexing. The authors have chosen Pnma and Cmcm for the low and high T phase. Although this is usually the one reported, this is very much inconvenient since high and low T phase do not have the same setting and cannot be compared directly. This makes the reading very difficult and the comparison of the high and low T phase data somehow cumbersome (X zone boundary become Y etc..). I strongly suggest that the authors use a consistent set of two space groups (possibly with a translation). Ccmm should be compatible with the Pnma setting.

In fact the Figure 1 shows the Cmcm and Pnma diffraction pattern with the same indexing convention (Pmna?). It seems that the same convention is taken through the all paper but sometime X ZB becomes Y ZB so that it is confusing. This point should be clarified from the beginning of the paper. The three unit cell parameter should be given.

- A more detailed description of the low and high T phase has to be provided. Indeed it is quite difficult to grasp atomic changes from the Figure 1 of the paper. Moreover, as explained in the paper and previously published work, there is a large Bragg peak intensity redistribution which plays a major role in the transition. In particular the symmetry changes, the symmetry element loss and their explanation should be given in the SI part.

- The Figure 2 is very interesting but confusing. One would expect the panel e-h to be the high T part of a-e, but this is not the case. The two panels illustrates two different branches better visible due to the S(Q,E) selection rules. The TA and TO should be shown at east in one panel with arrows to facilitate the reading. Again here it seems that the Pnma indexing is used.

Why is it written: (at $H = 2$ in panel h for Cmcm corresponding to $H = 1$ in panel d for Pnma)?

2- Some other remarks or comments:

2-1 Visualisation of S(Q,E) intensity map :

The susceptibility is interesting but a normalisation by $E^n(E)$ is in general preferable, since in that case the acoustic mode have a constant intensity. The S(Q,E) expression contains this $1/E$ term that can be cancelled.

Indeed some plots are difficult to read at low q because the signal is dominated by a rapidly varying acoustic intensity (see fig 1 SI)

2.2 Figure 2 : Why does the simulation display an intensity distribution in a wide range of Energy at low q ? One would expect well-defined acoustic excitations.

2.3 SI Figure 3 : Is the temperature correct in panel k and l ? It seems to be a higher T.

Why no data along H01 are show at low T?

2.4 'As T is increased above T_C , the TAc branch stiffens back up along its entirety (see also SFig. 3 q-t).'

This stiffening is not really visible. In fact the linear regime remains extremely limited as compared to the Pnma structure. One may wonder what is the structural origin of this remaining instability up to high T. Discussion on the structure derived at high T in particular in term of adp, disorder etc... should

be carried out and added in the manuscript.

2.5 'At 150K the TAc branch (< 4 meV) shows greatest intensity closest to (102), whereas at 860K intensity is highest near (002). In SFig. 3 we also observe drastic changes in the relative intensities of

the TOc mode at (103) and (203) (panels a-e) and (002) and (102) (panels f-j).'

It seems that the effect is mainly due to the Bragg peak intensity redistribution through the phase transition, not so much the anharmonicity. A simple calculation could be done to check this point. For TA the integrated $S(Q,E)$ intensity is proportional to the Bragg peak intensity. For TO one may suggest that the mode intensity is inversely proportional to the Bragg peak structure factor (ie strong when the Bragg peak is weak, which seems to be the case.)

2.6 Phonon lifetime.

The manuscript presents very nice experimental and simulated phonon lifetimes. A qualitative comparison on a few points between simulation and experiment could have been provided.

2.7 SnSe diffraction pattern and intensity distribution should be provided in the SI for both high and low T phases. .

Authors' response to Reviewers' comments

Reviewer #1 (Remarks to the Author):

Recommendation: minor revision

Comments:

Tin sulfide (SnS) is a type of layer structure material with strong anharmonicity, and aroused much attention for its promising energy conversion applications. This work reported the phase structure transition progress of SnS and further studied its phonon dispersions by experimental measurements and theoretical calculations. The investigation of atomic scale structure evolution uncovered the phonon scattering and soft-mode mechanism during phase transition. It is a comprehensive and solid work for understanding the phonon transports in materials with structural phase transition. This paper is well written, the experimental results and theoretical calculations are self-consistent, and the discussion is reasonable. Here are some comments and suggestions should be considered as below.

1. From inelastic neutron scattering (INS) experiments, the authors observed striking “softening” behaviors of TA and TO modes before phase transition (Pnma), however the TA branch stiffens up rather than softening with increasing temperature at Cmcm phase region. Therefore, what's the difference of this temperature-dependent behavior for TA mode in Pnma and Cmcm? And what effects will that have on the phonon propagation or thermal transport during phase transition?

We thank the reviewer for appreciating the “striking” nature of the TA and TO mode softening in the Pnma upon warming, for $T < T_c$. The reviewer raises a good point that, indeed, the behaviors are somewhat ‘inverted’ in the Pnma and Cmcm phase. In fact, in both phases the soft mode (TO for Pnma and zone-boundary TA for Cmcm) softens upon approaching T_c . But one phase is below T_c (Pnma), so the TO mode softens on warming ($T < T_c$), whereas the other phase is above T_c (Cmcm), so there the TA mode softens on cooling ($T > T_c$). This behavior is actually characteristic of soft mode transitions in general: the mode associated with the transition should soften as T reaches T_c from either side. Hence, the stiffening upon warming for $T > T_c$ is not unexpected. The reviewer can consult a classic example of a soft-mode transition in the following reference, from which the figure to the right is reproduced (soft-mode energy ΔE in lower panel):

[Redacted]

G. Shirane and Y. Yamada, “Lattice-Dynamical Study of the 110°K Phase Transition in SrTiO₃”, Phys. Rev. 177, 858 (1969)

We note that in SrTiO₃, upon cooling the high-temperature cubic phase to near $T_c \sim 110$ K, a zone-boundary mode (R point of cubic Brillouin

Figure reproduced from : G. Shirane and Y. Yamada, “Lattice-Dynamical Study of the 110°K Phase Transition in SrTiO₃”, Phys. Rev. 177, 858 (1969)

zone) becomes soft, i.e. its energy reaches zero. This creates a distorted tetragonal unit cell, twice as large, and thus leads to new superstructure Bragg peaks at the R-point wave-vectors of the cubic phase. The soft-mode re-emerges and stiffens on further cooling for $T < T_c$. This is similar in trends to SnS, except at a much lower T_c , and with different crystal structures.

The stiffening upon warming in the Cmcm phase ($T > T_c$) could in principle lead to an increase in group velocities, which would act to increase thermal conductivity (see manuscript Eqn. 1). However, this mode is very flat (even after some stiffening) and hence does not directly contribute much to the thermal conductivity. The stronger effect is likely on the scattering phase space. Further, the Cmcm phase is only stable over a limited temperature range above the transition, so the full stiffening of phonons at the highest temperatures is intrinsically difficult to access.

Several sentences in the text have been improved to clarify these points to the reader: “The soft mode behavior can be seen on both sides of T_c in the INS data (CTAX), since these are not limited to the zone-center since no modes are forbidden due to symmetry as in Raman. In both phases the soft mode (TO_c for Pnma and zone-boundary TA_c for Cmcm) softens upon approaching T_c . This behavior is characteristic of soft mode transitions in general and has been observed in seminal studies of phase transitions such as the work of Shirane et al. on $SrTiO_3$ [30].”

2. I am a bit of confusion that what does the sharp increment of linewidths mean near the phase transition in Fig. 3d, does it relate to the abrupt change of thermal lattice conductivity corresponding to T_c ?

We thank the referee for this question. As detailed in the manuscript, the drastic renormalization of the phonon dispersions, linked to the soft-mode behavior, causes an increase in phonon scattering rates and linewidths. Very close to T_c , the phonons are strongly overdamped (as shown in Fig. 3). This type of divergent behavior near a second order (continuous) phase transition can be ascribed to so-called critical behavior, similar to the divergence in the heat capacity (shown in supplement), which is also inversely observed in the thermal diffusivity (thus a significant cancellation of the divergence occurs in the thermal conductivity).

We have now clarified this point in the sentence “This divergence behavior near a second order (continuous) phase transition can be ascribed to critical behavior, similar to the divergence in the heat capacity (see SFig.17).” in the manuscript (page 4).

3. In the first-principles thermal conductivity simulations, how the authors estimate the contribution of acoustic and optical branches, and their temperature-dependent group velocities? In addition, the authors have calculated the temperature dependence of lattice conductivity before 800 K, how about the thermal conductivity near phase transition (from Pnma to Cmcm)?

The reviewer is asking good questions, since some studies in the literature (for example on the related SnSe) have made crude approximations to model the thermal conductivity. In comparison, in our study, *we include all phonon branches to compute the lattice thermal conductivity*, according to Eq.1 in the main text:

$$\kappa_{lat} = \sum_{q,s} C_{v,q,s} v_{g,q,s} \otimes v_{g,q,s} \tau_{q,s},$$

where C_v is specific heat, v_g is group velocity and τ is phonon lifetime. The double sum is over all wavevectors (q) and branches (s). Since κ_{lat} is calculated by summing over phonon modes it is relatively

straightforward to separate out the contributions from the acoustic and optic modes. However, it is important to note that the scattering rates for different modes are coupled through the scattering phase space. Further, we do explicitly consider the effects of “temperature-dependent group velocities” (renormalization), via the combination of AIMD and TDEP simulations. We have in particular performed a k_{lat} calculation close to T_c , as asked by the reviewer, (at $T=846\text{K}$) and the results are shown in Fig.4-c and Table 1. These calculations were performed using renormalized second order force constants ($\phi^{(2)}$) from TDEP at 846K and third order force constants ($\phi^{(3)}$) calculated from almaBTE. At temperatures very near T_c , divergences of physical properties associated with critical behavior could make simulations less accurate, and the notion of phonon quasiparticle breaks down entirely for the modes that become overdamped, since their frequency is no longer well defined. Thus, we preferred not to attempt k_{lat} calculations very near, or across, T_c .

4. In the discussion part of anharmonic renormalization, how the authors realize the phonon renormalization with temperature to simulate the thermal conductivity? How does this renormalization effect lattice conductivity?

This is a valid question and we are keen to clarify this point. As discussed in the text (Methods section), the effects of renormalization are applied through the temperature-dependent effective potentials (TDEP) method [7] which extracts renormalized force-constants from temperature-dependent AIMD simulations through the fitting of the potential energy surface. The phonon renormalization affects k_{lat} in three ways by:

1. Directly altering the group velocities ($v_g = |\nabla_{\mathbf{k}}\omega|$),
2. Changing energy-dependent phonon occupations (through the Bose-Einstein factor $n(E,T)$),
3. Changing the phonon lifetimes by changing the phase-space for scattering (itself a function of all dispersions).

The last two points affect the scattering rates as defined in Eq.3 of the main text. On pages 5-7 of the manuscript we provide a detailed discussion of how the renormalization affects k_{lat} , showing that the dominant effect is from the increase in scattering rates.

Reviewer #2 (Remarks to the Author)

Review report for the manuscript titled, "Extended anharmonic collapse of phonon dispersions in SnS" by T. Lanigan-Atkins et al.

The authors have performed extensive INS and High resolution Raman measurements to demonstrate the phase transition mechanism in SnS from Pnma phase to Cmcm phase. The authors have measured the thermal conductivity of SnS and compared with first principles calculations using TDEP/AlmaBTE codes. The manuscript is very clearly written and figures are informative. I have a few questions/comments which need to be addressed before I recommend this manuscript for publication in Nat. Comm.

We thank Reviewer #2 for appreciating our "extensive INS and High resolution Raman measurements" and for finding the manuscript "very clearly written".

Major comments:

1. The authors claim that, in fig. 3a, the Raman peak disappears after the phase transition to Cmcm phase. But in fig. 2e, there seems to be a zone center optic phonon mode (at Gamma) in a similar energy scale as in fig. 3a. So unless that optic mode in fig. 2e is not Raman active, I would expect that mode to show up in the Raman spectrum. Can the authors explain/comment on this point?

First, we wish to clarify the reciprocal space notations and evolution of phonon dispersions shown in Figure 2. In Fig 2a, the reciprocal space path (Pnma notation) is from one zone center (Gamma) to another zone center (Gamma', corresponding to a superlattice Bragg peak only present in the Pnma phase), through the X point. This same path, in the parent Cmcm phase, corresponds to zone-center (Gamma) to the zone-boundary (Y) point. The Y point is the location of the soft-mode that gives rise to the distortion and superstructure Gamma' zone center in Pnma. Thus the mode at ~1meV at Y in Fig 2e is not a zone-center mode. It is the only mode that is at comparable energy as the zone-center Raman active TO mode in Pnma at high temperature in Fig 3a, but because it is not at the Cmcm zone-center, it is not Raman active. On the other hand, the mode at ~4.5 meV at Gamma in the Cmcm phase is also not Raman active, as explained below.

The high temperature phase of SnS has the Cmcm space-group with atoms occupying the 4c Wyckoff positions. Using the Symmetry Adapted Modes [1] program from the Bilbao Crystallographic Server we find that for the high temperature phase of SnS/SnSe, only the following modes are Raman active:

Raman Active Modes

WP	A _g	A _u	B _{1g}	B _{1u}	B _{2g}	B _{2u}	B _{3g}	B _{3u}
4c	1	.	1	.	.	.	1	.

where '1' and '.' indicate that the mode is Raman active and inactive respectively. Our calculations show the lowest energy optic mode (at Gamma) in the Cmcm phase to belong to the B_{2g} representation. Hence, we expect this mode to be Raman inactive which is what we observe in our measurements.

2. Can the authors give a physical reasoning as to why phonon renormalization from TDEP increases the scattering rates compared to the 0 K force constants (see fig. 4e)? As far as I know, the renormalization reduces the interactions among phonons (see Cowley's paper: R A Cowley 1968 Rep. Prog. Phys. 31 123 and Wallace's book: Thermodynamics of Crystals).

The Reviewer is asking a good question (although we are not sure which exact part of these two lengthy references the Reviewer has in mind). The phonon scattering rates in Equation 3 (proportional to phonon linewidths measured in spectroscopy experiments) depend on the phonon dispersions (frequencies) and thus on phonon renormalization, in two ways:

1. Phonon energy softening increases the phonon thermal occupation for a given T_{BE} .
2. The three-phonon scattering phase space is modulated as the dispersions are shifted by renormalization. In particular, as the TO branch softens on warming in Pnma, the numbers of phonon interactions involving this mode increase.

The phonon linewidths shown in Table 1 below are for the lowest-energy TO mode extracted from the same calculations as in main text Figure 4.e. As can be seen in this table, comparing the cases of 0K harmonic $\Phi^{(2)}$ with 600K TDEP $\Phi^{(2)}$ for the same T_{BE} , we see that the effect of $\Phi^{(2)}$ renormalization on the TO phonon linewidth at fixed T_{BE} is quite large, resulting in a factor of five larger linewidth at either $T_{BE}= 300K$ or $T_{BE}= 846K$.

Table 1. Phonon linewidth of TO mode at $q=(0.2, 0, 0)$ from harmonic approximation and from TDEP with $\Phi^{(3)}$ from AlmaBTE, at various T_{BE} .

$\Phi^{(2)}$	T_{BE}	Linewidth (meV)
0K harmonic	300K	0.18
0K harmonic	846K	0.51
600K TDEP	300K	0.90
600K TDEP	846K	2.60

The increase in linewidth of the lowest-energy TO phonon mode is measured and explained in the main text: "The Γ_c linewidth shows a drastic increase at the transition (Fig. 3-d), reaching a value 9.4× higher than at 295 K, corresponding to a drastically increased scattering rate. Importantly, this increase is far larger than the linear temperature dependence expected within low-order perturbation theory ($n(E, T) \propto T$ at high T)."

The parameters affected by $\Phi^{(2)}$ are described in the main text: "Group velocities directly depend on the phonon frequencies, which can be obtained from either the harmonic or renormalized phonon models. In addition, scattering rates Γ_s for different phonon branches s also depend on energies (Eq. 2 [41]), although more indirectly, through the scattering phase space and the Bose-Einstein occupation factor (Eq. 3)."

3. The authors have essentially used the comparison between two calculations (one with 0 K, harmonic Φ^2 and one with TDEP Φ^2 at 600 K, both with TDEP Φ^3) to conclude that the differences in the scattering rates rather than the group velocities (upon renormalization) has the dominant “renormalization effect” on the thermal conductivity. I don’t think this is a fair comparison. Using 0 K Φ^2 and TDEP@600 K Φ^3 will lead to a completely different free energy manifold than the TDEP Φ^2 and Φ^3 at 600 K. In order to draw this conclusion, a fair comparison would be between 0 K Φ^2 and 0 K Φ^3 (like the supercell displacement method implemented in AlmaBTE) and TDEP Φ^2 and TDEP Φ^3 .

We thank the reviewer for giving us a chance to clarify the motivation and details for the specific calculation in the main text.

First, we want to clarify the calculations we presented in the original manuscript. For the thermal conductivity calculations in the original submission, we did not use TDEP Φ^3 . Rather, the Φ^3 were obtained from the supercell displacement method implemented in AlmaBTE (effectively 0K Φ^3). In addition, the same 0K Φ^3 from AlmaBTE were used for all of the thermal conductivity calculations in the main text.

The motivation for our approach was the following: we sought to isolate the contribution to the thermal conductivity from the Φ^2 renormalization only, and thus we kept the Φ^3 values fixed. The main text explains why we changed Φ^2 while fixing Φ^3 :

“To investigate intrinsic renormalization effects, Φ^3 were combined with Φ^2 from either harmonic 0K or 600 K TDEP calculations. These Φ^2 are then used to calculate κ at different T_{BE} . The strong reduction in κ upon changing Φ^2 is found to be dominated by changes in phonon lifetimes rather than group velocities, reflecting changes in the phase space from the pronounced phonon renormalization. To establish this, we calculate the change in κ from changing Φ^2 calculated at 0 K (harmonic) to those obtained at 600 K (AIMD-TDEP), whilst fixing $T_{BE} = 300$ K (Table I, first versus third row and Fig. 4-f blue curve versus red curve).”

To be clearer, we changed the main text to: “To investigate intrinsic renormalization effects, Φ^3 were combined with Φ^2 from either harmonic 0K or 600 K TDEP calculations. These Φ^2 are then used to calculate κ at different T_{BE} with fixed Φ^3 ”

Further, we feel that what the reviewer perhaps considers the combination of almaBTE and TDEP methodologies to be “unfair”. To address this concern, we have now also performed calculations of thermal conductivities and phonon scattering rates fully from TDEP.

The response Figure 1 below shows the thermal conductivity calculated with TDEP (Φ^2 and Φ^3 from 300K AIMD) at fixed $T_{BE} = 300$ K (using BTE), with increasing phonon q-points density. As seen in Figure 1, the thermal conductivity is fairly well converged with a grid of 15x15x15 q points (using the adaptive Gaussian integration method). We note that this is the largest grid size we could run on computers we have access to. Our new TDEP calculations result in a somewhat better agreement with our measurements than our previous almaBTE+TDEP calculations, although the trends are very similar (see response Fig. 2 below and revised manuscript Fig. 4c). Most importantly, these new calculations also show that our previous conclusions regarding the respective importance of group velocities vs scattering rates still bear out (increased scattering rates is the dominant effect).

The details of our TDEP thermal conductivity calculations have been added to the Methods section:

“Third-order force constants were also extracted with the TDEP method from the AIMD trajectories at 300K and 600K with an interaction cutoff distance $r = 6.12 \text{ \AA}$. The TDEP software was then used to calculate lattice thermal conductivity with a $15 \times 15 \times 15$ phonon q-point mesh. The thermal conductivities are converged with a grid of $15 \times 15 \times 15$ q points, with $\Phi^{(2)}$ and $\Phi^{(3)}$ from TDEP at 300K. In TDEP, the thermal conductivities are calculated using the adaptive Gaussian integration method and solving the BTE equation iteratively at $T_{BE} = 423\text{K}$ and $T_{BE} = 846 \text{ K}$.”

Figure 1 Convergence test of q-point mesh for TDEP thermal conductivity calculations. Thermal conductivity calculations were done with TDEP and using $\Phi^{(2)}$ and $\Phi^{(3)}$ from 300K AIMD extracted by TDEP. The thermal conductivity is calculated with adaptive Gaussian integration, at $T_{BE} = 300 \text{ K}$.

The TDEP thermal conductivity calculations have now been added to the main text and compared with experimental thermal conductivity in Fig. 4c, and the following text was added:

“ κ_{lat} calculated from TDEP with $\Phi^{(2)}$ and $\Phi^{(3)}$ from AIMD are shown in Fig. 4-c, and details of TDEP thermal conductivity calculations are in Methods. κ_{lat} calculated from TDEP with $\Phi^{(2)}$ and $\Phi^{(3)}$ from AIMD agree better with experimental results comparing with κ_{lat} calculated with fixed $\Phi^{(3)}$ from AlmaBTE.”

Figure 2 Measured κ (circles) and calculated κ_{lat} (lines and markers) along the a, b and c crystallographic directions. Calculated κ_{lat} (lines) uses $\Phi^{(2)}$ from 0K DFT with almaBTE while triangles are from TDEP $\Phi^{(2)}$. Stars are κ_{lat} obtained with almaBTE but with renormalized $\Phi^{(2)}$ from TDEP.

Based on our additional TDEP calculations, we confirmed our previously stated conclusions that the dominant effect to thermal conductivity is from the renormalization-induced changes in the phonon lifetimes. Table 2 below presents TDEP calculations confirming that the changes in thermal conductivities are mainly from changes in phonon lifetimes rather than group velocities. In all cases in Table 2, we fixed the value of T_{BE} to 300K. Rows 1 and 4 show the TDEP thermal conductivity calculated using TDEP force-constants from AIMD at 300K and 600K, respectively. From 300K to 600K, κ_{lat} decreases from 2.10W/mK to 1.77W/mK, (a 16% decrease). Rows 1 and 2 compare the thermal conductivities obtained from different group velocities (300K TDEP vs 600K TDEP), and show a slight increase in thermal conductivity (from 2.10W/mK to 2.24W/mK), which is due to the increasing average group velocity from 1040.9m/s to 1116.9m/s, respectively. Comparing rows 1 and 3, the phonon lifetimes are changed from 300K TDEP to 600K TDEP, and the thermal conductivity decreases from 2.10W/mK to 1.59W/mK, a 24% suppression over a limited T range. Thus, our new TDEP simulations further support our conclusion that the change in scattering rates has the dominant “renormalization effect”, compared to the more minor effect of group velocity renormalization.

Table 2 Lattice thermal conductivities calculated from TDEP with RTA, affected by changing phonon lifetime and phonon group velocity.

Index	Phonon dispersion	T_{BE}	Group velocity	Phonon lifetime	Average thermal conductivity (W/m/K)	Percentage
1	300K TDEP	300K	300K TDEP	300K TDEP	2.10	100%
2	300K TDEP	300K	600K TDEP	300K TDEP	2.24	107%
3	300K TDEP	300K	300K TDEP	600K TDEP	1.59	76%
4	600K TDEP	300K	600K TDEP	600K TDEP	1.77	84%

The following text was added to the main text (p. 7):

“Using TDEP, we computed κ_{lat} with renormalized force-constants ($\Phi_{(2)}$ and $\Phi_{(3)}$) from AIMD at 300 K and 600 K, with fixed $T_{BE} = 300$ K to isolate renormalization effects. Renormalizing both $\Phi_{(2)}$ and $\Phi_{(3)}$ suppresses κ_{lat} from $2.10 \text{ Wm}^{-1}\text{K}^{-1}$ to $1.77 \text{ Wm}^{-1}\text{K}^{-1}$, (a 16% decrease). Including only the effect of renormalized group velocities would actually increase κ_{lat} from $2.10 \text{ Wm}^{-1}\text{K}^{-1}$ to $2.24 \text{ Wm}^{-1}\text{K}^{-1}$, owing to the increased average group velocity (from 1041 ms^{-1} to 1117 ms^{-1}). On the other hand, changing the phonon lifetimes from 300K TDEP to 600K TDEP results in a thermal conductivity suppression from $2.10 \text{ Wm}^{-1}\text{K}^{-1}$ to $1.59 \text{ Wm}^{-1}\text{K}^{-1}$, a 24% suppression. Thus, our TDEP simulations further support our conclusion that the change in scattering rates is the dominant renormalization effect, compared to the more minor effect of group velocity renormalization. ”

4. Does the free energy calculation in TDEP also predict that the Pnma phase is the stable phase below $\sim T_c$ and Cmcm phase is the stable phase above T_c ? Further, what is the theoretical T_c predicted by TDEP?

This is a difficult question. Our study does not aim to accurately predict thermodynamic phase stability of either the Pnma or Cmcm structures fully from first-principles. However, we can comment on the predicted dynamic stability (positivity of dynamical matrix eigenvalues). The phonon dispersions of Pnma in harmonic approximation are stable, meanwhile, phonon dispersion extracted with TDEP from AIMD for the Pnma phase at 300K, 400K, 500K and 600K are stable as shown in SFig.8. However, phonon dispersions for Cmcm phase are not stable in the harmonic approximation. But the phonon dispersions for the Cmcm phase extracted with TDEP at 550K, 600K, 700K, 800K and 900K are stable and shown in supplement SFig.9. Based on this, one could venture that the TDEP method would predict a phase coexistence, similar to alternate anharmonic renormalization methods applied to SnSe [19].

The condensation temperature of the soft-mode in the Pnma phase was estimated as illustrated supplement Fig.10, using a linear interpolation of $\Phi^{(2)}$ between 0K DFT and 600K TDEP. The estimated condensation temperature $T_c = 624\text{K}$ is obtained by fitting the energy of TO soft mode with $E(T) = A * |T$

– $T_C |^\alpha$. In main text Fig. 3, this critical temperature is plotted on a shifted temperature axis (top axis) to align with the experimental phase transition temperature (880K).

As was further described in the main text (p. 5), “A linear interpolation between the harmonic and renormalized $\Phi^{(2)}$ was subsequently used to approximate the temperature dependence of $\Phi^{(2)}$ in the Pnma phase. The resulting temperature dependence of the TOc mode could be fitted with the same power law as above, resulting in an estimated transition temperature $T_C = 624\text{K}$ and $\alpha = 0.28$, the latter of which is in reasonable agreement with experimental results. The underestimation of T_C is consistent with recent anharmonic simulations on SnSe [19]. Simulation temperatures were scaled to match the experimental $T_C = 880\text{K}$, therefore 600 K is scaled to 846 K when comparing with experiments.”

Minor comments:

1. Why did the authors decide to use AlmaBTE for the BE solution and not the BE solver in TDEP itself?

For the thermal conductivity analysis, we wanted to isolate the effect from the second order $\Phi^{(2)}$ only, thus we needed to fix the $\Phi^{(3)}$ and keep all other parameters of the calculation identical. Comparing calculations performed with different codes can be complicated by different implementations, differences in convergence, etc. We thus elected to use AlmaBTE to determine the effect of changing $\Phi^{(2)}$ from the 0K harmonic values to the TDEP renormalized values at 300K and 600K, while the $\Phi^{(3)}$ were kept fixed. We also note that almaBTE is widely used for thermal conductivity calculations, and has been widely tested by the community.

However, to satisfy the reviewer, we have now added new TDEP thermal conductivity calculations, which result in similar trends and validate our previous conclusions.

2. Did the authors solve the BE iteratively for the thermal conductivity? Is there a big difference between the Relaxation Time Approximation and the full solution of the BE for the thermal conductivity?

We used AlmaBTE to perform the thermal conductivity calculation, where BTE calculations are initialized based on RTA approximation. The RTA and iterative BTE solutions from almaBTE are provided in the table below. As can be seen, the results are very close.

T_{BE}	$\Phi^{(2)}$	BTE average k_{latt} ($\text{Wm}^{-1}\text{K}^{-1}$)	RTA average k_{latt} ($\text{Wm}^{-1}\text{K}^{-1}$)
300K	DFT	2.25	2.13
846K	AIMD 600K	0.49	0.47

3. Please define DHO in the main text. It is only defined in the Methods section first.

DHO stands for “damped harmonic oscillator”. This has been updated in the manuscript.

“Phonon lineshapes were successfully modeled as damped harmonic oscillators (**DHO**, see Supp. Eq. 5) which approximates to a Lorentzian lineshape in the usual case where the phonon energy E_0 is much larger than the damping Γ_{LW} ” (page 4).

Reviewer #3 (Remarks to the Author):

This is a very nice paper on lattice dynamics as a function of temperature of SnS and SnSe.

The paper presents an impressive set of new inelastic neutron scattering phonon measurements, with a detailed study of the temperature dependence of the dispersion but also of the intensity distribution $S(Q,E)$ a crucial parameter when comparing with simulations.

A spectacular evolution of the $S(Q,E)$ response function is observed near the phase transition, with a phonon softening on a very large portion of the Brillouin zone for specific propagation vectors.

Results are compared to state of the art atomic scale simulation using DFT but also energy renormalisation through AIMD and TDEP. The agreement between experiment and simulation is striking.

The effect of the phonon renormalisation on the lattice dynamics has also been studied in details: the strong reduction of the lattice thermal conductivity is mainly due to a strong decrease of phonon lifetime due to renormalisation, whereas the group velocity renormalisation does not play a major role.

The paper is of a broad interest, and brings in important results both in the field of soft mode phase transition and for the understanding of their low thermal conductivity. The paper certainly deserves to be published in Nature Communication.

We thank the reviewer #3 for appreciating our “impressive set of new inelastic neutron scattering phonon measurements”, the “spectacular evolution of the $S(Q,E)$ ” that we report and the “striking agreement” with our “state of the art atomic scale simulations”.

However the authors should answer the following remarks and provide some changes in their manuscript.

1- The first remark concern the readability of the paper. The paper gathered an impressive set of both experimental and simulation results, as illustrated by the large supplementary information file. However I have the feeling that reading from a non specialist is sometime a bit difficult.

I made the following suggestions :

- Structure and space groups, indexing. The authors have chosen Pnma and Cmcm for the low and high T phase. Although this is usually the one reported, this is very much inconvenient since high and low T phase do not have the same setting and cannot be compared directly. This makes the reading very difficult and the comparison of the high and low T phase data somehow cumbersome (X zone boundary become Y etc..). I strongly suggest that the authors use a consistent set of two space groups (possibly with a translation). Cmmm should be compatible with the Pnma setting.

In fact the Figure 1 shows the Cmcm and Pnma diffraction pattern with the same indexing convention (Pmna?). It seems that the same convention is taken through the all paper but sometime X ZB becomes Y ZB so that it is confusing. This point should be clarified from the beginning of the paper. The three unit cell parameter should be given.

We thank the reviewer for the comments concerning improving the readability of the paper, particularly regarding the notation used. We have strongly considered using Pbnm for the low-temperature phase, which is an alternate setting for Pnma (space group 62), but decided against this as we wish to facilitate the comparison of our results with the relevant literature, in which Pnma is most widely used. Similarly,

the high temperature phase could be described as Bbmm (space group 63), but all the recent literature uses the Cmcm notation for the high temperature phase, including references [8-11].

However, we hope that the reviewer will appreciate that we have taken several steps to clarify the notation and the crystallography:

- Fig.1: We have edited the reciprocal space maps substantially to clarify the correspondence between the conventional Pnma and primitive Cmcm cells. Specifically, in panels b and f the Miller indices have been updated to include the Cmcm primitive notation in parentheses in smaller gray text. Labels were added for the (302) and (-101) (Pnma notation) reflections. Further, these reciprocal space maps have been rotated (90deg clockwise) to match the orientation of the real-space crystal structures shown in panels a and e. The caption was updated (as indicated by blue text) to clarify the notation. Further, the conventional Pnma and primitive Cmcm Brillouin zones have been added to b and f, respectively, together with high-symmetry points X and Y. As the reviewer can see, the superlattice peaks in the low-temperature Pnma phase occur at the zone-boundary of the Cmcm primitive Brillouin zone (Y point at $(\frac{1}{2}, \frac{1}{2}, 0)$). In panel e the lattice vectors are now shown for the primitive Cmcm unit cell.

- Fig.2: We have edited all panels to show Cmcm notation in parentheses and have changed the caption to make clearer which notation we are using.

- A more detailed description of the low and high T phase has to be provided. Indeed it is quite difficult to grasp atomic changes from the Figure 1 of the paper. Moreover, as explained in the paper and previously published work, there is a large Bragg peak intensity redistribution which plays a major role in the transition. In particular the symmetry changes, the symmetry element loss and their explanation should be given in the SI part.

The description of the phase transition in the main manuscript has been expanded and a discussion of the symmetry changes associated with the Pnma-Cmcm transition has been added to the supplement section B. We have also revised and improved Figure 1 to clarify the notations and the correspondence between the two structures in both real and reciprocal space. We note that the crystallography is discussed extensively in the literature we cited, and this does not constitute the main focus of our study, which is primarily concerned with the lattice dynamics (soft mode and connection with thermal transport).

- The Figure 2 is very interesting but confusing. One would expect the panel e-h to be the high T part of a-e, but this is not the case. The two panels illustrates two different branches better visible due to the S(Q,E) selection rules. The TA and TO should be shown at least in one panel with arrows to facilitate the reading. Again here it seems that the Pnma indexing is used.

Why is it written: (at H = 2 in panel h for Cmcm corresponding to H = 1 in panel d for pi314etc

We thank reviewer 3 for recognizing Fig.2 as a striking figure. To measure the extremely low-energy TA mode in the Cmcm phase it was necessary to employ high resolution neutron scattering using cold neutrons. The low-energy neutrons introduce kinematic restraints that meant it was not possible to measure the same [H02] direction measured for the Pnma phase (outside the accessible Q-range). However, we stress that the directions [H02] and [H01] are exactly equivalent in the irreducible Brillouin zone and the polarization condition is very similar such that we are measuring c-polarized modes with

reduced q-vector along [100] in both cases. Panels a-d and e-h are for different phases so it is not possible to have exactly equivalent branches.

We agree with the reviewer that labeled arrows can greatly simplify the communication of the results. We have now added arrows to Fig.2-b,f indicating the TA and TO modes. Further, we have also added an extra figure to the supplement (SFig.4) which shows more clearly the effect of mode softening and structure evolution on the scattering intensity. This figure is also included below (Fig.4) for convenience. Panels a-b used the internal atomic coordinates (r_d) from the relaxed Pnma structure whereas c uses r_d from the relaxed Cmcm structure (since internal coordinates in Pnma phase near T_c are nearly the same as those in Cmcm phase). The shifts in atomic coordinates cause a large redistribution in intensity along branches: in panels a-b the acoustic mode close to $H = 1$ is strongest but in panel c, $H = 0$ gives the strongest acoustic intensity (following the evolution of Bragg peak intensities). This is borne out in the experimental measurements (d-f). Note: the broader lineshapes in c, compared to a-b, are the result of applying a different resolution function since we wish to compare panel c with the data measured on HB3 (shown in panel f), whereas data shown in panels d-e were measured on CNCS. The text “at $H = 2$ in panel h for Cmcm corresponding to $H = 1$ in panel d for Pnma” has been revised to “at $H = 2$ in panel h for Cmcm and $H = 1$ in panel d for Pnma which are both superstructure peaks”

Figure 3: Temperature evolution of transverse phonon dispersions in SnS along [100] across the structural phase transition. Simulated $\chi''(\mathbf{Q}, E)$ from harmonic second-order force constants (a) and renormalized second-order force constants from AIMD (b-c) show the c-polarized TA and TO modes as indicated. Panels (d-e) show experimental $\chi''(\mathbf{Q}, E)$ measured on CNCS whilst (f) is from HB-3. Panels (a-b) use internal coordinates from the relaxed Pnma structure whereas c uses r_d from the relaxed Cmcm structure.

2- Some other remarks or comments:

2-1 Visualisation of $S(\mathbf{Q},E)$ intensity map :

The susceptibility is interesting but a normalisation by $E^n(E)$ is in general preferable, since in that case the acoustic mode have a constant intensity. The $S(\mathbf{Q},E)$ expression contains this $1/E$ term that can be cancelled.

Indeed some plots are difficult to read at low q because the signal is dominated by a rapidly varying acoustic intensity (see fig 1 SI)

We thank the reviewer for alerting us to the fact that SFig.1 was mislabeled as $\chi''(\mathbf{Q},E)$ when it was in fact $S(\mathbf{Q},E)$. We have now included plots for both $\chi''(\mathbf{Q},E)$ (SFig.2) and $S(\mathbf{Q},E)$ (SFig.1) in the supplement. The $\chi''(\mathbf{Q},E)$ plot makes it much easier to compare phonons at different energies.

2.2 Figure 2 : Why does the simulation display an intensity distribution in a wide range of Energy at low q ? One would expect well-defined acoustic excitations.

The dispersions are obtained from the second-order force-constants and so as the reviewer intimated should be delta functions in energy. However, our intensity simulations also take into account both instrumental resolution and the effect of integrating in \mathbf{Q} to match the experimental \mathbf{Q} -integration. These effects lead to a distribution in energy. However, thanks to the Reviewer's question, we realized that the \mathbf{Q} -integration range for Fig. 2b was inadvertently larger than the experiment. We thank the reviewer for pointing this out and we have now corrected it.

2.3 SI Figure 3 : Is the temperature correct in panel k and l ? It seems to be a higher T.

Why no data along H01 are show at low T?

The temperature labels are correct. The difference between f-j and k-o comes from the instruments the data were collected on. The data presented in panels f-j were measured on CNCS, which uses cold neutrons to achieve a high energy resolution, whereas data in k-o were measured with the thermal triple-axis spectrometer HB-3, which has poorer energy resolution. Note: when extracting linewidths these differences were accounted for to obtain intrinsic phonon linewidths.

We have added plots showing data measured with CNCS along [H01] at 300K and 667K to SFig.4. Due to time constraints, we were not able to cover the same angular range at all temperatures on CNCS, thus we only have two temperatures for [H01].

Phonon intensities are much stronger along [H02] than along [H01], due to the \mathbf{Q} -dependence of the structure factor, and hence we chose to focus on showing the better quality data measured along [H02]. The panels p-t show data measured on CTAX, which uses very low energy neutrons to achieve very high resolution. Due to the coupling of \mathbf{Q} and energy in neutron measurements this meant that we were unable to access the [H02] direction for these high-resolution measurements.

2.4 'As T is increased above T_C , the TAc branch stiffens back up along its entirety (see also SFig. 3 q-t).'

This stiffening is not really visible. In fact the linear regime remains extremely limited as compared to the Pnma structure. One may wonder what is the structural origin of this remaining instability up to high T. Discussion on the structure derived at high T in particular in term of adp, disorder etc... should be carried out and added in the manuscript.

Figure 3-b shows spectra for the soft mode, including above T_c (zone boundary acoustic mode in Cmcm phase) at T=960K (E = 1.14±0.02 meV) and 1050K (E = 1.40±0.02 meV, a 23% increase). We hope that the reviewer can see the clear upward shift (stiffening) on heating on this figure. Measurements at additional temperatures are summarized in Fig 3-c, showing the stiffening trend. Fig.2-g,f also shows that the stiffening is extending between H=1.5 and H=2.

The high-temperature Cmcm structure, including atomic displacement parameters (ADPs) / thermal disorder, is extensively discussed in the literature we cited, for example the following references:

[11] T. Chattopadhyay, J. Pannetier, and H. Von Schner- ing, Journal of Physics and Chemistry of Solids 47, 879 (1986).

[14] K. Adouby, C. Perez-Vicente, and J. Jumas, Z. Kristal- logr 213, 343 (1998).

2.5 'At 150K the TAc branch (< 4 meV) shows greatest intensity closest to (102), whereas at 860K intensity is highest near (002). In SFig. 3 we also observe drastic changes in the relative intensities of the TOc mode at (103) and (203) (panels a-e) and (002) and (102) (panels f-j).'

It seems that the effect is mainly due to the Bragg peak intensity redistribution through the phase transition, not so much the anharmonicity. A simple calculation could be done to check this point. For TA the integrated S(Q,E) intensity is proportional to the Bragg peak intensity. For TO one may suggest that the mode intensity is inversely proportional to the Bragg peak structure factor (ie strong when the Bragg peak is weak, which seems to be the case.)

The reviewer is correct that large changes in dynamic structure factor result from the evolution of internal coordinates, which we have investigated in Fig.4 below, which has also been added to the Supplement (new figure SFig.5). To clarify this point, we compared two simulations of $\chi(\mathbf{Q},E)$, both using renormalized second-order force constants extracted from 600K AIMD (846K after scaling) using TDEP, with experimental data measured at 860K on HB3. Panel (a) uses internal coordinates (r_d) from the relaxed theoretical Pnma structure whereas (b) uses r_d from the relaxed theoretical Cmcm structure (experimentally, at high T, the internal coordinates of the Pnma phase tend towards those of Cmcm as expected for continuous phase transitions). Clearly, the intensity distribution is very different between (a) and (b), which results purely from the rearrangement of the relative atomic positions. Further, panel (b) uses Cmcm r_d and gives a better match to the experimental results. From this figure we can clearly say that, indeed, r_d has a big impact on S(Q,E). Yet, the reviewer can also appreciate that the intensity redistribution is decoupled from the softening of the branches (temperature dependence of mode frequencies), as was shown in Fig. 3 above.

Further, there is clearly also a large anharmonic effect from the phonon eigenvectors changing with temperature. We can see this by comparing Fig.3-a to Fig.4-a where the only difference is that the former uses $\Phi^{(2)}$ from 0K DFT whilst the latter uses renormalized $\Phi^{(2)}$ extracted from 600K AIMD (846K after

scaling) using TDEP. In Fig.3-a the acoustic modes are strongest close to $H = 1$ whereas in Fig.4-a they are strongest close to $H = 0$. The trends are reversed for optical modes.

From these observations, it is apparent that the drastic changes in relative intensities arise from a combination of shifts in atomic positions with temperature and anharmonic renormalization.

Figure 4: As SnS changes from Pnma to Cmcm there is a continuous evolution of internal coordinates (r_d) which impacts the $\chi''(Q, E)$ intensities. Here we show $\chi''(Q, E)$ calculated using renormalized second-order force constants for the Pnma phase from 600K AIMD (864K after scaling) with internal coordinates from 0K DFT for the Pnma phase (a) and for the Cmcm phase (b).. Panel (c) shows experimental data from HB-3.

2.6 Phonon lifetime.

The manuscript present very nice experimental and simulated phonon lifetimes. A qualitative comparison on a fw point between simulation and experiment could have been provided.

We thank the reviewer for appreciating our “very nice experimental and simulated phonon lifetimes”. We followed the reviewer’s suggestion, and we have now added in the supplement a table directly comparing the linewidth of several additional phonon modes. The phonon linewidths shown in response Table 3 below directly compares linewidths from first-principles calculations and from Raman measurements.

Table 3 Phonon linewidth of zone-center Raman-active modes from almaBTE (with $\Phi^{(2)}$) from 0K harmonic or TDEP calculation, and $\Phi^{(3)}$ from AlmaBTE), at various T_{BE} .

$\Phi^{(2)}$	T_{BE}	TO_c Linewidth (meV)	Ag^1 Linewidth (meV)	B_{3g}^0 Linewidth (meV)
0K harmonic	300K	0.21 (0.15±0.03 @ 293K)	0.38 (0.33±0.04 @ 293K)	0.12 (0.16±0.03 @ 293K)
0K harmonic	600K	0.43	0.74	0.24
0K harmonic	846K	0.60	1.05	0.34
600K TDEP	846K	3.33 (1.03 ± 0.18 @ 853K)	2.06 (1.84±0.16 @ 844K)	0.38 (0.43±0.20 @ 696K)

2.7 SnSe diffraction pattern and intensity distribution should be provided in the SI for both high and low T phases.

The diffraction intensities for SnSe measured on HB-3 are shown in Fig.5 below and in SFig.29. These scans are from the same experiment as the S(Q,E) data shown in S Figs.1-2. Our study was focused on the lattice dynamics and hence we undertook primarily inelastic scattering. We were not focused on elastic scattering and did not map out full diffraction patterns as the instruments are not optimized for this. However, for SnS we did perform elastic scans for the peaks measured on HB-3 and CTAX (Fig. 1-c,g) where we clearly observed the superstructure peaks vanishing in the Cmcm phase. We have also added elastic maps (S Fig. 28) showing the (HOL) plane for SnS as measured on CNCS for five temperatures in the Pnma phase.

There are numerous detailed diffraction studies of SnSe and SnS in the literature which are cited in the manuscript and reproduced here for convenience [2-5].

Figure 5: Diffraction intensities for SnSe as measured on HB-3 at ORNL

List of all changes in the manuscript and supplemental document

Text:

All changes made to the text are highlighted in blue in the manuscript and supplemental document. A new section (B) has been added to the supplemental document to describe the symmetry changes across the phase transition as suggested by Reviewer 3. All captions have been updated to reflect that all log intensity color maps are in \log_{10} .

Add to the main text according to Reviewer 2 explain thermal conductivity from TDEP:

κ_{lat} calculated from TDEP with $\Phi^{(2)}$ and $\Phi^{(3)}$ from AIMD are shown in Fig. 4-c, and details of TDEP thermal conductivity calculations are in Methods. κ_{lat} calculated from TDEP with $\Phi^{(2)}$ and $\Phi^{(3)}$ from AIMD agree better with experimental results comparing with κ_{lat} calculated with fixed $\Phi^{(3)}$ from AlmaBTE.

Thermal conductivities from 300K to 600K TDEP $\Phi^{(2)}$ and $\Phi^{(3)}$ at $T_{BE} = 300$ K decrease from $2.10 \text{ Wm}^{-1} \text{ K}^{-1}$ to $1.77 \text{ Wm}^{-1} \text{ K}^{-1}$, (a 16% decrease). Compare the thermal conductivities obtained from different group velocities (300K TDEP vs 600K TDEP), thermal conductivity increases from $2.10 \text{ Wm}^{-1} \text{ K}^{-1}$ to $2.24 \text{ Wm}^{-1} \text{ K}^{-1}$, which is due to the increasing average group velocity from 1040.9 m/s to 1116.9 m/s for 300K TDEP and 600K TDEP respectively. When the phonon lifetimes are changed from 300K TDEP to 600K TDEP, and the thermal conductivity decreases from $2.10 \text{ Wm}^{-1} \text{ K}^{-1}$ to $1.59 \text{ Wm}^{-1} \text{ K}^{-1}$, a 24% suppression. Thus, our TDEP simulations further support our conclusion that the change in scattering rates has the dominant renormalization effect, compared to the more minor effect of group velocity renormalization.

Third order force constants were also extracted with TDEP method from 300K and 600K AIMD at an interaction cutoff distance $r = 6.12 \text{ \AA}$. Second order force constants (described above) and third order force constants from TDEP are used to calculate lattice thermal conductivity with phonon q-point mesh $15 \times 15 \times 15$. The thermal conductivities are converged with a grid of $15 \times 15 \times 15$ q points, with $\Phi^{(2)}$ and $\Phi^{(3)}$ from TDEP at $T_{BE} = 300$ K (see SFig. 14). In TDEP, the thermal conductivities are calculated using the adaptive Gaussian integration method and solving the BTE equation iteratively at $T_{BE} = 423$ K and $T_{BE} = 846$ K.

Add to the main text according to Reviewer 2 to address the thermal conductivity calculations we did: These $\Phi^{(2)}$ are then used to calculate κ at different T_{BE} with fixed $\Phi^{(3)}$.

Figures:

Fig.1 - In panels b and f the Miller indices have been updated to include the Cmc21 primitive notation in parentheses in smaller gray text. Labels were added for the (302) (Pnma notation) reflection. Further, these reciprocal space maps have been rotated (90deg clockwise) to match the orientation of the real-space crystal structures shown in panels a and e. The caption was updated (as indicated by blue text) to clarify the notation. The conventional Pnma and primitive Cmc21 Brillouin zones have been added to b and f, respectively. In panel e the lattice vectors are now shown for the primitive Cmc21 unit cell.

Fig.2 - Arrows have been added to panels b and f to indicate the TA and TO modes. The labelling of the x-axes for all panels has been updated to show both Pnma notation (no parentheses) and Cmc21 (in parentheses). The tick marks have been updated on all of the colormaps to be clearer. The caption was updated (as indicated by blue text) to further clarify the notation.

Fig.3 - The Raman energies and linewidths have been updated in panels c and d because our fitting has been improved. Previously we were using an instrumental resolution (FWHM = 0.21 meV) which was slightly too large. We have now updated our fitting to use an instrument resolution (FWHM = 0.09 meV), measured on the same system and same configuration, obtained by fitting the laser line to a Gaussian and accounting for the intrinsic laser linewidth ($1.7 \times 10^{-4} \text{ cm}^{-1}$)

Fig.4 - Panel c was updated to include the new lattice thermal conductivity calculations using $\Phi^{(2)}$ and $\Phi^{(3)}$ both from TDEP (shown as stars) as described in the response to Reviewer 2's major comment 3. The caption to Fig.4 has been updated to reflect the changes in panel c.

SFig.1 - Reviewer 3 comment 2-1 alerted us to the fact that this figure was previously mislabelled as $\chi''(\mathbf{Q}, E)$ whereas it was actually showing $S(\mathbf{Q}, E)$. This has since been corrected in the caption.

SFig.2 - As suggested by Reviewer 3, $\text{Chi}''(\mathbf{Q},E)$ can be used to better distinguish the low-energy portions of acoustic dispersions. Hence we added this figure which is showing the same data as SFig.1 but expressed as $\text{Chi}''(\mathbf{Q},E)$.

SFig.3 (formerly SFig.2) - The data from Duke has been updated with the new instrumental resolution parameters as described for Fig.3.

SFig.4 - This is a new figure used to help show the gradual changes in the TO_c and TA_c phonon energies and relative intensities along [100]. This was added in response to Reviewer 3 comment 1 to help explain the changes in dispersions more clearly.

SFig.5 - This a new figure that was added to show the effects of altering the internal coordinates (rd) of the unit cell on the intensity ($\text{Chi}''(\mathbf{Q},E)$) in response to Reviewer 3 comment 2.5.

SFig.6 (formerly SFig.3) - Panels u and v were added as requested by Reviewer 3 comment 2.3 and the caption has been updated accordingly.

SFig.12 (formerly SFig.9) - The caption has been corrected to $\text{Chi}''(\mathbf{Q},E)$ whereas previously it was incorrectly described as $S(\mathbf{Q},E)$.

SFig.14 - This is a new figure demonstrating the q-mesh convergence of the new lattice thermal conductivity calculations using both $\Phi^{(2)}$ and $\Phi^{(3)}$ from TDEP which were performed in response to Reviewer 2's comments.

SFig.19 (formerly SFig.15) - The Raman linewidths have been updated from the new fitting. Previously we were using an instrumental resolution (1.65cm^{-1}) which was slightly too large. We have now updated our fitting to use an instrument resolution (0.7cm^{-1}), measured on the same system and same configuration, obtained by fitting the laser line to a Gaussian and accounting for the intrinsic laser linewidth ($1.7 \times 10^{-4}\text{cm}^{-1}$).

SFig.20 (formerly SFig.16) - The Raman energies have been updated from the new fitting. Previously we were using an instrumental resolution (1.65cm^{-1}) which was slightly too large. We have now updated our fitting to use an instrument resolution (0.7cm^{-1}), measured on the same system and same configuration, obtained by fitting the laser line to a Gaussian and accounting for the intrinsic laser linewidth ($1.7 \times 10^{-4}\text{cm}^{-1}$). Any changes to the energies were extremely minor.

SFig.30 - This is a new figure showing the diffraction results for SnSe as a function of temperature when it was measured on HB-3. This data/figure was requested by Reviewer 3 comment 2.7.

SFig.31 - This is a new figure showing the diffraction results for SnS, in the (H0L) plane, as a function of temperature when it was measured on CNCS. This data/figure was added in response to Reviewer 3 comment 2.7.

Tables:

Table 1 Supplement - This table comparing linewidths from simulations with those from Raman measurements was added in response to Reviewer 3 comment 2.6.

References:

1. Kroumova *et. al.* Phase Transitions (2003), 76, Nos. 1-2, 155-170.

2. T. Chattopadhyay, A. Werner, H. Von-Schnering, and J. Pannetier, *Revue de Physique Appliquee* 19, 807 (1984).
3. T. Chattopadhyay, J. Pannetier, and H. Von Schnering, *Journal of Physics and Chemistry of Solids* 47, 879 (1986).
4. K. Adouby, C. Perez-Vicente, and J. Jumas, *Z. Kristallogr* 213, 343 (1998).
5. H. G. Von Schnering and H. Wiedemeier, *Zeitschrift fur Kristallographie-Crystalline Materials* 156, 143 (1981).
6. Agne, Matthias T., Peter W. Voorhees, and G. Jeffrey Snyder. "Phase Transformation contributions to heat capacity and impact on thermal diffusivity, thermal conductivity, and thermoelectric performance." *Advanced Materials* 31.35 (2019): 1902980.
7. Hellman, Olle, and Igor A. Abrikosov. "Temperature-dependent effective third-order interatomic force constants from first principles." *Physical Review B* 88.14 (2013): 144301.
8. Aseginolaza, Unai, et al. "Strong anharmonicity and high thermoelectric efficiency in high-temperature SnS from first principles." *Physical Review B* 100.21 (2019): 214307.
9. Dewandre, Antoine, et al. "Two-step phase transition in SnSe and the origins of its high power factor from first principles." *Physical review letters* 117.27 (2016): 276601.
10. Chatterji, Tapan, et al. "Soft-phonon dynamics of the thermoelectric β -SnSe at high temperatures." *Physics Letters A* 382.29 (2018): 1937-1941.
11. Lee, C-H., et al. "Extremely space and time restricted thermal transport in the high temperature Cmcm phase of thermoelectric SnSe." *Materials Today Physics* (2019): 100171.
12. Carrete, Jesus, et al. "almaBTE: A solver of the space–time dependent Boltzmann transport equation for phonons in structured materials." *Computer Physics Communications* 220 (2017): 351-362.

REVIEWER COMMENTS

Reviewer #1 (Remarks to the Author):

authors well addressed all my comments, it can be published as is.

Reviewer #2 (Remarks to the Author):

Review report for the manuscript titled, "Extended anharmonic collapse of phonon dispersions in SnS" by T. Lanigan-Atkins et al.

I thank the authors for carefully addressing my comments. I have one minor follow-up concern, which needs to be addressed. I recommend publication of the manuscript provided the authors address it.

The authors give two reasons for renormalization increasing the scattering rates. While it is a plausible explanation, I would like to see more quantitative evidence. My concern is that, in fig. 4(e), the scattering rates for almost all of the phonons increase upon TDEP renormalization, with some of them (around 2 meV phonon energy) being more than 10 times larger for the case with renormalization compared to the case without renormalization. So, plots of detailed three-phonon phase space or any other quantity that can definitively explain this unusual observation would be useful.

Reviewer #3 (Remarks to the Author):

The authors have answered all remarks and modified the manuscript and supplemental information file accordingly.

The manuscript can be accepted in the present form.

Authors response

Reviewer #1 (Remarks to the Author):

authors well addressed all my comments, it can be published as is.

We thank reviewer 1 for recommending publication as-is.

Reviewer #2 (Remarks to the Author):

Review report for the manuscript titled, "Extended anharmonic collapse of phonon dispersions in SnS" by T. Lanigan-Atkins et al.

I thank the authors for carefully addressing my comments. I have one minor follow-up concern, which needs to be addressed. I recommend publication of the manuscript provided the authors address it.

The authors give two reasons for renormalization increasing the scattering rates. While it is a plausible explanation, I would like to see more quantitative evidence. My concern is that, in fig. 4(e), the scattering rates for almost all of the phonons increase upon TDEP renormalization, with some of them (around 2 meV phonon energy) being more than 10 times larger for the case with renormalization compared to the case without renormalization. So, plots of detailed three-phonon phase space or any other quantity that can definitively explain this unusual observation would be useful.

Reviewer 2 is asking for additional computations in this second round of review. These are very specialized, technical first-principles calculations, as would typically be found in purely theoretical studies. We wish to emphasize that our paper already contains a huge amount of results, with a primary focus on state-of-the-art inelastic neutron scattering measurements revealing the observation of a drastic softening of optical and acoustic phonons in SnS (and related SnSe) at high temperature, and a discussion of the connection with thermal transport.

However, as requested by Reviewer 2, we have now performed additional calculations of the three-phonon scattering phase-space. These calculations further confirm the conclusions in our manuscript. We performed the first-principles calculations of the three-phonon weighted scattering phase-space, as defined in reference [1] equation (3), and as implemented in the software AlmaBTE (ShengBTE emulator mode):

$$W_{\lambda}^{\pm} = \frac{1}{2N} \sum_{\lambda' p''} \left\{ \begin{array}{l} 2(f_{\lambda'} - f_{\lambda''}) \\ f_{\lambda'} + f_{\lambda''} + 1 \end{array} \right\} \frac{\delta(\omega_{\lambda} \pm \omega_{\lambda'} - \omega_{\lambda''})}{\omega_{\lambda} \omega_{\lambda'} \omega_{\lambda''}}, \quad (3)$$

As pointed out by Reviewer 2, some SnS phonon modes at low energy exhibit a large increase in linewidths (scattering rates) upon dispersion renormalization. Specifically, as shown in Fig 4(e) of our manuscript, some phonon branches with energy between 1meV to 3meV are strongly impacted. First, we have now identified the nature of these modes. We find that the most strongly impacted (large linewidths upon dispersion renormalization) are the soft TO branch near the zone-center (along different directions) and some long-wavelength acoustic modes, especially TA

branches. This is perhaps intuitive since these modes are strongly impacted by the TO soft mode energy renormalization.

Further, we computed the three-phonon weighted scattering phase-space for both the harmonic dispersions (0K) and the renormalized dispersions (TDEP at 600K), both with the same phonon occupation temperature (846K). We tabulate below results characteristic of the two types of modes (TO and TA, both near Gamma) in Table 1 below.

Table 1. Phonon energy renormalization and change in three-phonon scattering phase-space for characteristic modes strongly impacted by dispersion renormalization in manuscript Fig. 4-e

600K	qa (RLU)	qb (RLU)	qc (RLU)	E (meV)	Weighted phase space (ps ⁴ /rad ⁴)
TO	0.000	0.000	0.000	2.028	0.128
TA	0.000	0.091	0.091	1.815	0.102
0K	qa (RLU)	qb (RLU)	qc (RLU)	E (meV)	Weighted phase space (ps ⁴ /rad ⁴)
TO	0.000	0.000	0.000	5.036	0.036
TA	0.000	0.091	0.091	2.159	0.086

As can be seen in this table, dispersion renormalization (energy softening from 0K to 600K) results in a significant increase in weighted phase-space for phonon-phonon scattering (almost four-fold increase for zone-center TO mode and 20% for this particular TA mode). Other similar modes (long-wavelength TA and TO modes) follow similar trends. Thus, our manuscript statement that dispersion renormalization increases phonon scattering via the increase in phase-space is validated. Further, we also maintain that renormalization also suppresses group velocities, and increases the phonon occupations at high temperature. All three effects of dispersion renormalization are combined to suppress the thermal conductivity at high temperature in SnS.

[1] Wu Li and Natalio Mingo, "Ultralow lattice thermal conductivity of the fully filled skutterudite YbFe₄Sb₁₂ due to the flat avoided-crossing filler modes", Phys. Rev. B 91, 144304 (2015)

Reviewer #3 (Remarks to the Author):

The authors have answered all remarks and modified the manuscript and supplemental information file accordingly.

The manuscript can be accepted in the present form.

We thank reviewer 3 for recommending acceptance of our manuscript in its present form.

Summary of changes to the manuscript:

We have now added a statement to our manuscript [an increase in scattering rates (linewidths), shown in 4-e, "especially pronounced for low-energy TO and TA modes below 5meV"] to further qualify that sentence on page 7.

REVIEWERS' COMMENTS:

Reviewer #2 (Remarks to the Author):

The authors have satisfactorily addressed my concern. I recommend the publication of this manuscript in its current form.